# RAMP: Boosting Adversarial Robustness Against Multiple $l_p$ Perturbations for Universal Robustness

**Enyi Jiang**
Department of Computer Science
University of Illinois Urbana-Champaign
Urbana, IL 61801
enyij2@illinois.edu

**Gagandeep Singh**
Department of Computer Science
University of Illinois Urbana-Champaign
Urbana, IL 61801
ggnds@illinois.edu

## Abstract

Most existing works focus on improving robustness against adversarial attacks bounded by a single $l_p$ norm using adversarial training (AT). However, these AT models' multiple-norm robustness (union accuracy) is still low, which is crucial since in the real-world an adversary is not necessarily bounded by a single norm. The tradeoffs among robustness against multiple $l_p$ perturbations and accuracy/robustness make obtaining good union and clean accuracy challenging. We design a logit pairing loss to improve the union accuracy by analyzing the tradeoffs from the lens of distribution shifts. We connect natural training (NT) with AT via gradient projection, to incorporate useful information from NT into AT, where we empirically and theoretically show it moderates the accuracy/robustness tradeoff. We propose a novel training framework **RAMP**, to boost the robustness against multiple $l_p$ perturbations. **RAMP** can be easily adapted for robust fine-tuning and full AT. For robust fine-tuning, **RAMP** obtains a union accuracy up to $53.3\%$ on CIFAR-10, and $29.1\%$ on ImageNet. For training from scratch, **RAMP** achieves a union accuracy of $44.6\%$ and good clean accuracy of $81.2\%$ on ResNet-18 against AutoAttack on CIFAR-10. Beyond multi-norm robustness **RAMP**-trained models achieve superior *universal robustness*, effectively generalizing against a range of unseen adversaries and natural corruptions.

## 1 Introduction

Though deep neural networks (DNNs) demonstrate superior performance in various vision applications, they are vulnerable against adversarial examples [Goodfellow et al., 2014, Kurakin et al., 2018]. Adversarial training (AT) [Tramèr et al., 2017, Madry et al., 2017] which works by injecting adversarial examples into training for enhanced robustness, is currently the most popular defense. However, most AT methods address only a *single* type of perturbation [Wang et al., 2020, Wu et al., 2020, Carmon et al., 2019, Gowal et al., 2020, Raghunathan et al., 2020, Zhang et al., 2021, Debenedetti and Troncoso—EPFL, 2022, Peng et al., 2023, Wang et al., 2023]. An $l_\infty$ robust model may not be robust against $l_p(p \neq \infty)$ attacks. Also, enhancing robustness against one perturbation type can sometimes increase vulnerability to others [Engstrom et al., 2017, Schott et al., 2018]. On the contrary, training a model to be robust against multiple $l_p$ perturbations is crucial as it reflects real-world scenarios [Sharif et al., 2016, Eykholt et al., 2018, Song et al., 2018, Athalye et al., 2018] where adversaries can use multiple $l_p$ perturbations. We show that multi-norm robustness is the key to improving generalization against other threat models [Croce and Hein, 2022]. For instance, we show it enables robustness against perturbations not easily defined mathematically, such as image corruptions and unseen adversaries [Wong and Kolter, 2020].

38th Conference on Neural Information Processing Systems (NeurIPS 2024).

Two main challenges exist for training models robust against multiple perturbations: (i) tradeoff among robustness against different perturbation models [Tramer and Boneh, 2019] and (ii) tradeoff between accuracy and robustness [Zhang et al., 2019, Raghunathan et al., 2020]. Adversarial examples induce a shift from the original distribution, causing a drop in clean accuracy with AT [Xie et al., 2020, Benz et al., 2021]. The distinct distributions created by $l_1, l_2, l_\infty$ adversarial examples make the problem even more challenging. Through a finer analysis of the distribution shifts caused by these adversaries, we propose the **RAMP** framework to efficiently boost the **R**obustness **A**gainst **M**ultiple **P**erturbations. **RAMP** can be used for both fine-tuning and training from scratch. It utilizes a novel logit pairing loss on a certain pair and connects NT with AT via gradient projection [Jiang et al., 2023] to improve union accuracy while maintaining good clean accuracy and training efficiency.

**Logit pairing loss.** We visualize the changing of $l_1, l_2, l_\infty$ robustness when fine-tuning a $l_\infty$-AT pre-trained model in Figure 1 using the CIFAR-10 training dataset. The DNN loses substantial robustness against $l_\infty$ attack after only 1 epoch of fine-tuning: $l_1$ fine-tuning and E-AT [Croce and Hein, 2022] (red and yellow histograms under Linf category) both lose significant $l_\infty$ robustness (compared with blue histogram under Linf category). Inspired by this observation, we devise a new logit pairing loss for a $l_q - l_r$ tradeoff pair to attain better union accuracy, which enforces the logit distributions of $l_q$ and $l_r$ adversarial examples to be close, specifically on the correctly classified $l_q$ subsets. In comparison, our method (green histogram under Linf and union categories) preserves more $l_\infty$ and union robustness than others after 1 epoch. We show this technique works on larger models and datasets (Section 5.1).

**Connect natural training (NT) with AT.** We explore the connections between NT and AT to obtain a better accuracy/robustness trade-off. We find that NT can help with adversarial robustness: useful information in natural distribution can be extracted and leveraged to achieve better robustness. To this end, we compare the similarities of model updates of NT and AT *layer-wise* for each epoch, where we find and incorporate useful NT components into AT via gradient projection (GP), as outlined in Algorithm 2. In Figure 2 and Section 5.1, we empirically and theoretically show this technique strikes a better balance between accuracy and robustness, for both single and multiple $l_p$ perturbations. We provide a theoretical analysis of why GP works for adversarial robustness in Theorem A.2 & 4.5.

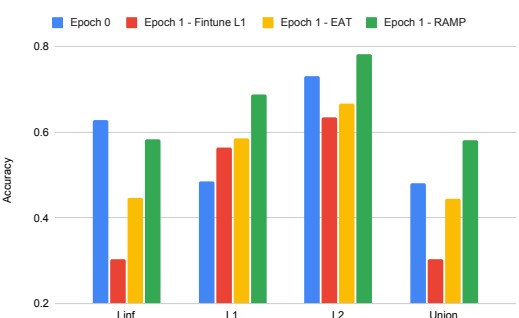

Figure 1: **Multiple-norm tradeoff with robust fine-tuning**: We observe that fine-tuning on $l_\infty$-AT model using $l_1$ examples drastically reduces $l_\infty$ robustness. **RAMP** preserves more $l_\infty$ and union robustness.

**Main contributions**:

- We design a new logit pairing loss to mitigate the $l_q - l_r$ tradeoff for better union accuracy, by enforcing the logit distributions of $l_q$ and $l_r$ adversarial examples to be close.

- We empirically and theoretically show that connecting NT with AT via gradient projection better balances the accuracy/robustness tradeoff for $l_p$ perturbations, compared with standard AT.

- **RAMP** achieves good union accuracy, accuracy-robustness tradeoff, and generalizes better to diverse perturbations and corruptions (Section 5.1) achieving superior *universal robustness* (75.5% for common corruption and 26.1% union accuracy against unseen adversaries). **RAMP** fine-tuned DNNs achieve union accuracy up to 53.3% on CIFAR-10, and 29.1% on ImageNet. **RAMP** achieves a 44.6% union accuracy and good clean accuracy on ResNet-18 against AutoAttack on CIFAR-10. Our code is available at `https://github.com/uiuc-focal-lab/RAMP`.

## 2 Related Work

**Adversarial training (AT).** Adversarial Training (AT) usually employs gradient descent to discover adversarial examples, incorporating them into training for enhanced adversarial robustness [Tramèr et al., 2017, Madry et al., 2017]. Numerous works focus on improving robustness by exploring the trade-off between robustness and accuracy [Zhang et al., 2019, Wang et al., 2020], instance reweighting [Zhang et al., 2021], loss landscapes [Wu et al., 2020], wider/larger architectures [Gowal

et al., 2020, Debenedetti and Troncoso—EPFL, 2022], data augmentation [Carmon et al., 2019, Raghunathan et al., 2020], and using synthetic data [Peng et al., 2023, Wang et al., 2023]. However, these methods often yield DNNs robust against a *single* perturbation type while remaining vulnerable to other types.

**Robustness against multiple perturbations.** Tramer and Boneh [2019], Kang et al. [2019] observe that robustness against $l_p$ attacks does not necessarily transfer to other $l_q$ attacks ($q \neq p$). Previous studies [Tramer and Boneh, 2019, Maini et al., 2020, Madaan et al., 2021, Croce and Hein, 2022] modified Adversarial Training (AT) to enhance robustness against multiple $l_p$ attacks, employing average-case [Tramer and Boneh, 2019], worst-case [Tramer and Boneh, 2019, Maini et al., 2020], and random-sampled [Madaan et al., 2021, Croce and Hein, 2022] defenses. There are also works [Nandy et al., 2020, Liu et al., 2020, Xu et al., 2021, Xiao et al., 2022, Maini et al., 2022] using preprocessing, ensemble methods, mixture of experts, and stability analysis to solve this problem. Ensemble models and preprocessing methods are weakened since their performance heavily relies on correctly classifying or detecting various types of adversarial examples. In certified training, Banerjee et al. [2024], Banerjee and Singh [2024] propose verification/certifiable training methods under different threat models for $l_p$ universal adversarial perturbation. However, prior works are hard to scale to larger models and datasets, e.g. ImageNet, due to the efficiency issue. Furthermore, Croce and Hein [2022] devise Extreme norm Adversarial Training (E-AT) and fine-tune a $l_p$ robust model on another $l_q$ perturbation to quickly make a DNN robust against multiple $l_p$ attacks. However, E-AT does not adapt to varying epsilon values. Our work demonstrates that the suboptimal tradeoff observed in prior studies can be improved with our proposed framework.

**Logit pairing in adversarial training.** Adversarial logit pairing methods encourage logits for pairs of examples to be similar [Kannan et al., 2018, Engstrom et al., 2018]. People apply this technique to both clean images and their adversarial counterparts, to devise a stronger form of adversarial training. In our work, we devise a novel logit pairing loss to train a DNN originally robust against $l_p$ attack to become robust against another $l_q (q \neq p)$ attack on the correctly predicted $l_p$ subsets, which helps gain better union accuracy.

**Adversarial versus distributional robustness.** Sinha et al. [2018] theoretically studies the AT problem through distributional robust optimization. Mehrabi et al. [2021] establishes a pareto-optimal tradeoff between standard and adversarial risks by perturbing the test distribution. Other works explore the connection between natural and adversarial distribution shifts [Moayeri et al., 2022, Alhamoud et al., 2023], assessing transferability and generalizability of adversarial robustness across datasets. However, little research delves into distribution shifts induced by $l_1, l_2, l_\infty$ adversarial examples and their interplay with the robustness-accuracy tradeoff [Zhang et al., 2019, Yang et al., 2020, Rade and Moosavi-Dezfooli, 2021]. Our work, inspired by recent domain adaptation techniques [Jiang, 2023, Jiang et al., 2023], designs a logit pairing loss and utilizes model updates from NT via GP to enhance adversarial robustness. We show that GP adapts to both single and multi-norm scenarios.

## 3   AT against Multiple Perturbations

We consider a standard classification task with samples $\{(x_i, y_i)\}_{i=0}^N$ from an empirical data distribution $\widehat{\mathcal{D}}_n$; we have input images $x \in \mathbb{R}^d$ and corresponding labels $y \in \mathbb{R}^k$. Standard training aims to obtain a classifier $f$ parameterized by $\theta$ to minimize a loss function $\mathcal{L} : \mathbb{R}^k \times \mathbb{R}^k \to \mathbb{R}$ on $\widehat{\mathcal{D}}_n$. Adversarial training (AT) [Madry et al., 2017, Tramèr et al., 2017] aims to find a DNN robust against adversarial examples. It is framed as a min-max problem where a DNN is optimized using the worst-case examples within an adversarial region around each $x_i$. Different types of adversarial regions $B_p(x, \epsilon_p) = \{x' \in \mathbb{R}^d : \|x' - x\|_p \leq \epsilon_p\}$ can be defined around a given image $x$ using various $l_p$-based perturbations. Formally, we can write the optimization problem of AT against a certain $l_p$ attack as follows:

$$\min_\theta \mathbb{E}_{(x,y) \sim \widehat{\mathcal{D}}_n} \left[ \max_{x' \in B_p(x, \epsilon_p)} \mathcal{L}(f(x'), y) \right]$$

The above optimization is only for certain $p$ values and is usually vulnerable to other perturbation types. To this end, prior works have proposed several approaches to train the network robust against multiple perturbations ($l_1, l_2, l_\infty$) at the same time. We focus on the union threat model

$\Delta = B_1(x, \epsilon_1) \cup B_2(x, \epsilon_2) \cup B_\infty(x, \epsilon_\infty)$ which requires the DNN to be robust within the $l_1, l_2, l_\infty$ adversarial regions simultaneously [Croce and Hein, 2022]. Union accuracy is then defined as the robustness against $\Delta_{(i)}$ for each $x_i$ sampled from $\mathcal{D}$. In this paper, similar to the prior works, we use union accuracy as the main metric to evaluate the multiple-norm robustness. Apart from that, we define *universal robustness* as the generalization ability against a range of unseen adversaries and common corruptions. Specifically, we have average accuracy across five severity levels for common corruption and union accuracy against a range of unseen adversaries used in Laidlaw et al. [2020].

**Worst-case defense** follows the following min-max optimization problem to train DNNs using the worst-case example from the $l_1, l_2, l_\infty$ adversarial regions:

$$\min_\theta \mathbb{E}_{(x,y)\sim\widehat{\mathcal{D}}_n} \left[ \max_{p\in\{1,2,\infty\}} \max_{x'\in B_p(x,\epsilon_p)} \mathcal{L}(f(x'), y) \right]$$

MAX [Tramer and Boneh, 2019] and MSD [Maini et al., 2020] fall into this category. Finding worst-case examples yields a good union accuracy but results in a loss of clean accuracy as the distribution of generated examples is different from the clean data distribution.

**Average-case defense** train DNNs using the average of the $l_1, l_2, l_\infty$ worst-case examples:

$$\min_\theta \mathbb{E}_{(x,y)\sim\widehat{\mathcal{D}}_n} \left[ \mathbb{E}_{p\in\{1,2,\infty\}} \max_{x'\in B_p(x,\epsilon_p)} \mathcal{L}(f(x'), y) \right]$$

AVG [Tramer and Boneh, 2019] is of this type. This method generally leads to good clean accuracy but suboptimal union accuracy as it does not penalize worst-case behavior within the $l_1, l_2, l_\infty$ regions.

**Random-sampled defense.** The defenses mentioned above lead to a high training cost as they compute multiple attacks for each sample. SAT [Madaan et al., 2021] and E-AT [Croce and Hein, 2022] randomly sample one attack out of each type at a time, contributing to a similar computational cost as standard AT on a single perturbation model. They achieve a slightly better union accuracy compared with AVG and relatively good clean accuracy. However, they are not better than worst-case defenses for multiple-norm robustness, since they do not consider the strongest attack within the union region all the time.

## 4 RAMP

There are two main tradeoffs in achieving better union accuracy while maintaining good accuracy: 1. Among perturbations: there is a tradeoff among different attacks, e.g., a $l_\infty$ pre-trained AT DNN is not robust against $l_1, l_2$ perturbations, which makes the union accuracy harder to attain. Also, we observe there exists a main tradeoff pair of two attacks among the union over $l_1, l_2, l_\infty$ attacks. 2. Accuracy and robustness: all defenses lead to degraded clean accuracy. To address these tradeoffs, we study the problem from the lens of distribution shifts.

**Interpreting tradeoffs from the lens of distribution shifts.** The adversarial examples with respect to an empirical data distribution $\widehat{\mathcal{D}}_n$, adversarial region $B_p(x, \epsilon_p)$, and DNN $f_\theta$ generate a new adversarial distribution $\widehat{\mathcal{D}}_a$ with samples $\{(x'_i, y_i)\}_{i=0}^N$, that are correlated by adding certain perturbations but different from the original $\widehat{\mathcal{D}}_n$. Because of the shifts between $\widehat{\mathcal{D}}_n$ and $\widehat{\mathcal{D}}_a$, DNN decreases performance on $\widehat{\mathcal{D}}_n$ when we move away from it and towards $\widehat{\mathcal{D}}_a$. Also, the distinct distributions created by multiple perturbations, $\widehat{\mathcal{D}}_a^{l_1}, \widehat{\mathcal{D}}_a^{l_2}, \widehat{\mathcal{D}}_a^{l_\infty}$, contribute to the tradeoff among $l_1, l_2, l_\infty$ attacks. To address the tradeoff among perturbations while maintaining good efficiency, we focus on the distributional interconnections between $\widehat{\mathcal{D}}_n$ and $\widehat{\mathcal{D}}_a^{l_1}, \widehat{\mathcal{D}}_a^{l_2}, \widehat{\mathcal{D}}_a^{l_\infty}$. From the insights we get from above, we propose our framework **RAMP**, which includes (i) logit pairing to improve tradeoffs among multiple perturbations, and (ii) identifying and combining the useful DNN components using the model updates from NT and AT, to obtain a better robustness/accuracy tradeoff.

**Identify the Key Tradeoff Pair.** We study the common case with $l_p$ norms $\epsilon_1 = 12, \epsilon_2 = 0.5, \epsilon_\infty = \frac{8}{255}$ on CIFAR-10 [Tramer and Boneh, 2019]. The distributions generated by the two strongest attacks show the largest shifts from $\widehat{\mathcal{D}}_n$; also, they have the largest distribution shifts between each other

because of larger and most distinct search areas. Thus, by comparing the single norm robustness of $l_p$ adversarially trained models, we select the two $l_p$-AT models with the lowest $l_p$ robustness against themselves as the key tradeoff pair. They refer to the strongest attack since their $l_p$ robustness is low. The attack with the highest $l_p$ robustness is mostly included by the convex hull of the other two stronger attacks [Croce and Hein, 2022]. Here we identify $l_\infty - l_1$ as the key tradeoff pair.

## 4.1 Logit Pairing for Multiple Perturbations

**Figure 1: Finetuning a $l_q$-AT model on $l_r$ examples reduces $l_q$ robustness.** To get a finer analysis of the $l_\infty - l_1$ tradeoff mentioned above, we visualize the changing of $l_1, l_2, l_\infty$ robustness of the training dataset when we fine-tune a $l_\infty$ pre-trained model with $l_1$ examples for 1 epochs, as shown in Figure 1: x-axis represents the robustness against different attacks and y-axis is the accuracy. After 1 epoch of finetuning on $l_1$ examples or performing E-AT, we lose much $l_\infty$ robustness since blue/yellow histograms are much lower than the red histogram under the Linf category. RAMP preserves both $l_\infty$ and union robustness more effectively: the green histogram is higher than the red/yellow histogram under Linf and Union categories. Specifically, RAMP maintains 14%, 28% more union robustness than E-AT and $l_1$ fine-tuning. The above observations indicate the necessity of preserving more $l_q$ robustness as we adversarially fine-tune with $l_r$ adversarial examples on a $l_q$ pre-trained AT model, with $l_q - l_r$ as the key tradeoff pair, which inspires us to design our loss design with logit pairing. We want to enforce the *union predictions* between $l_q$ and $l_r(q \neq r)$ attacks: bringing the predictions of $l_q$ and $l_r(q \neq r)$ close to each other, specifically on the correctly predicted $l_q$ subsets. Based on our observations, we design a new logit pairing loss to enforce a DNN robust against one $l_q$ attack to be robust against another $l_r(q \neq r)$ attack.

**Enforcing the Union Prediction via Logit Pairing.** The $l_q - l_r(q \neq r)$ tradeoff leads us to the following principle to improve union accuracy: *for a given set of images, when we have a DNN robust against some $l_q$ examples, we want it to be robust against $l_r$ examples as well.* This serves as the main insight for our loss design: we want to *enforce* the logits predicted by $l_q$ and $l_r$ adversarial examples to be close, specifically on the correctly predicted $l_q$ subsets. To accomplish this, we design a KL-divergence (KL) loss between the predictions from $l_q$ and $l_r$ perturbations. For each batch of data $(x, y) \sim \mathcal{D}$, we generate $l_q$ and $l_r$ adversarial examples $x'_q, x'_r$ and their predictions $p_q, p_r$ using APGD [Croce and Hein, 2020]. Then, we select indices $\gamma$, which part elements of $p_q$ correctly predicts the ground truth $y$. We denote the size of the indices as $n_c$, and the batch size as $N$. We compute a KL-divergence loss over this set of samples using $KL(p_q[\gamma] \| p_r[\gamma])$ (Eq. 1). For the subset indexed by $\gamma$, we want to push its $l_r$ logit distribution towards its $l_q$ logit distribution, such that we prevent losing more $l_q$ robustness when training with $l_r$ adversarial examples.

$$\mathcal{L}_{KL} = \frac{1}{n_c} \cdot \sum_{i=1}^{n_c} \sum_{j=0}^{k} p_q[\gamma[i]][j] \cdot \log \left( \frac{p_q[\gamma[i]][j]}{p_r[\gamma[i]][j]} \right) \tag{1}$$

To further boost the union accuracy, apart from the KL loss, we add another loss term using a MAX-style approach in Eq. 2: we find the worst-case example between $l_q$ and $l_r$ adversarial regions by selecting the example with the higher loss. $\mathcal{L}_{max}$ is a cross-entropy loss over the approximated worst-case adversarial examples. Here, we use $\mathcal{L}_{ce}$ to represent the cross-entropy loss. Our final loss $\mathcal{L}$ combines $\mathcal{L}_{KL}$ and $\mathcal{L}_{max}$, via a hyper-parameter $\lambda$ in Eq. 3.

$$\mathcal{L}_{max} = \frac{1}{N} \sum_{i=0}^{N} \left[ \max_{p \in \{q,r\}} \max_{x'_i \in B_p(x, \epsilon_p)} \mathcal{L}_{ce}(f(x'_i), y_i) \right] \quad (2) \qquad \mathcal{L} = \mathcal{L}_{max} + \lambda \cdot \mathcal{L}_{KL} \tag{3}$$

Algorithm 1 shows the pseudocode of robust fine-tuning with **RAMP** that leverages logit pairing.

## 4.2 Connecting Natural Training with AT

To improve the robustness and accuracy tradeoff against multiple perturbations, we explore the connections between AT and NT. Since extracting valuable information in NT aids in improving robustness (Section 4.2), we use gradient projection [Jiang et al., 2023] to compare and integrate natural and adversarial model updates, which yields an improved tradeoff between robustness and accuracy.

**NT can help adversarial robustness.** Let us consider two models $f_1$ and $f_2$, where $f_1$ is randomly initialized and $f_2$ undergoes NT on $\widehat{\mathcal{D}}_n$ for $k$ epochs: $f_2$ results in a better *decision boundary* and higher clean accuracy. Performing AT on $f_1$ and $f_2$ subsequently, intuitively, $f_2$ becomes more robust than $f_1$ due to its improved decision boundary, leading to fewer misclassifications of adversarial examples. This effect is empirically shown in Figure 2. For **AT** (blue), standard AT against $l_\infty$ attack [Madry et al., 2017] is performed, while for **AT-pre** (red), 50 epochs of pre-training precede the standard AT procedure. **AT-pre** shows superior clean and robust accuracy on CIFAR-10 against $l_\infty$ PGD-20 attack with $\epsilon_\infty = 0.031$. Despite $\widehat{\mathcal{D}}_n$ and $\widehat{\mathcal{D}}_a$ are different, Figure 2 suggests valuable information in $\widehat{\mathcal{D}}_n$ that potentially enhances performance on $\widehat{\mathcal{D}}_a$.

**AT with Gradient Projection.** To connect NT with AT more effectively, we analyze the training procedures on $\widehat{\mathcal{D}}_n$ and $\widehat{\mathcal{D}}_a$. We consider model updates over all samples from $\widehat{\mathcal{D}}_n$ and $\widehat{\mathcal{D}}_a$, with the initial model $f^{(r)}$ at epoch $r$, and models $f_n^{(r)}$ and $f_a^{(r)}$ after 1 epoch of natural and adversarial training from the same starting point $f^{(r)}$, respectively. Here, we compare the natural updates $\widehat{g}_n = f_n^{(r)} - f^{(r)}$ and adversarial updates $\widehat{g}_a = f_a^{(r)} - f^{(r)}$. Due to distribution shift, an *angle* exists between them. Our goal is to identify useful components from $g_n$ and incorporate them into $g_a$ for increased robustness in $\widehat{\mathcal{D}}_a$ while maintaining accuracy in $\widehat{\mathcal{D}}_n$. Inspired by Jiang et al. [2023], we *layer-wisely* compute the cosine similarity between $\widehat{g}_n$ and $\widehat{g}_a$. For a specific layer $l$ of $\widehat{g}_n^l$ and $\widehat{g}_a^l$, we preserve a portion of $\widehat{g}_n^l$ based on their cosine similarity score (Eq.4). Negative scores indicate that $\widehat{g}_n^l$ is not beneficial for robustness in $\widehat{\mathcal{D}}_a$. Therefore, we filter components with similarity score $\leq 0$. We define the **GP** (Gradient Projection) operation in Eq.5 by projecting $\widehat{g}_a^l$ towards $\widehat{g}_n^l$.

$$\cos(\widehat{g}_n^l, \widehat{g}_a^l) = \frac{\widehat{g}_n^l \cdot \widehat{g}_a^l}{\|\widehat{g}_n^l\|\|\widehat{g}_a^l\|} \quad (4) \qquad \mathbf{GP}(\widehat{g}_n^l, \widehat{g}_a^l) = \begin{cases} \cos(\widehat{g}_n^l, \widehat{g}_a^l) \cdot \widehat{g}_n^l, & \cos(\widehat{g}_n^l, \widehat{g}_a^l) > 0 \\ 0, & \cos(\widehat{g}_n^l, \widehat{g}_a^l) \leq 0 \end{cases} \quad (5)$$

Therefore, the total projected (useful) model updates $g_p$ coming from $\widehat{g}_n$ could be computed as Eq. 6. We use $\mathcal{M}$ to denote all layers of the current model update. Note that $\bigcup_{l \in \mathcal{M}}$ concatenates all layers' useful natural model update components. A hyper-parameter $\beta$ is used to balance the contributions of $g_{GP}$ and $\widehat{g}_a$, as shown in Eq. 7. By finding a proper $\beta$ (0.5 as in Figure 4c), we can obtain better robustness on $\widehat{\mathcal{D}}_a$, as shown in Figure 2 and Figure 3. In Figure 2, with $\beta = 0.5$, **AT-GP** refers to AT with GP; for **AT-GP-pre**, we perform 50 epochs of NT before doing **AT-GP**. We see **AT-GP** obtains a better accuracy/robustness tradeoff than **AT**. We observe a similar trend for **AT-GP-pre** vs. **AT-pre**. Further, in Figure 3, **RN-18 $l_\infty$-GP** achieves good clean accuracy and better robustness than **RN-18 $l_\infty$** against AutoAttack [Croce and Hein, 2020].

$$g_p = \bigcup_{l \in \mathcal{M}} \mathbf{GP}(\widehat{g}_n^l, \widehat{g}_a^l) \quad (6) \qquad\qquad f^{(r+1)} = f^{(r)} + \beta \cdot g_p + (1 - \beta) \cdot \widehat{g}_a \quad (7)$$

---

**Algorithm 1** Fine-tuning via Logit Pairing

1: **Input:** model $f$, input samples $(x, y)$ from distribution $\widehat{\mathcal{D}}_n$, fine-tuning rounds $R$, hyper-parameter $\lambda$, adversarial regions $B_q, B_r$ with size $\epsilon_q$ and $\epsilon_r$, **APGD** attack.
2: **for** $r = 1, 2, ..., R$ **do**
3:    **for** $(x, y) \sim$ training set $\mathcal{D}$ **do**
4:      $x_q', p_q \leftarrow \mathbf{APGD}(B_q(x, \epsilon_q), y)$
5:      $x_r', p_r \leftarrow \mathbf{APGD}(B_r(x, \epsilon_r), y)$
6:      $\gamma \leftarrow where(argmax\, p_q = y)$
7:      $n_c \leftarrow \gamma.size()$
8:      calculate $\mathcal{L}$ using Eq. 3 and update $f$
9:    **end for**
10: **end for**
11: **Output:** model $f$.

**Algorithm 2** Connect AT with NT via GP

1: **Input:** model $f$, input images with distribution $\widehat{\mathcal{D}}_n$, training rounds $R$, adversarial region $B_p$ and its size $\epsilon_p$, $\beta$, natural training **NT** and adversarial training **AT**.
2: **for** $r = 1, 2, ..., R$ **do**
3:    $f_n \leftarrow \mathbf{NT}(f^{(r)}, \mathcal{D})$
4:    $f_a \leftarrow \mathbf{AT}(f^{(r)}, B_p, \epsilon_p, \mathcal{D})$
5:    compute $\widehat{g}_n \leftarrow f_n - f^{(r)}, \widehat{g}_a \leftarrow f_a - f^{(r)}$
6:    compute $g_p$ using Eq. 6
7:    update $f^{(r+1)}$ using Eq. 7 with $\beta$ and $\widehat{g}_a$
8: **end for**
9: **Output:** model $f$.

## 4.3 Theoretical Analysis of GP for Adversarial Robustness

We define $\mathcal{D}_n = \{(x_i, y_i)\}_{i=0}^{\infty}$ as the ideal data distribution with an infinite cardinality. Here, we consider a classifier $f_\theta$ at epoch $t$. We define $\mathcal{D}_a$ as the distribution created by $\{(x_i + \epsilon(f_\theta, x_i, y_i), y_i)\}_{i=0}^{\infty}$ where $(x_i, y_i) \sim \mathcal{D}_n$. $x_i + \epsilon(f_\theta, x_i, y_i)$ denotes the perturbed image, which could be both single and multiple perturbations based on $f_\theta$ itself.

**Assumption 4.1.** *We assume* $\widehat{\mathcal{D}}_n$ *consists of $N$ i.i.d. samples from the ideal distribution $\mathcal{D}_n$ and* $\widehat{\mathcal{D}}_a = \{(x_i + \epsilon(f^\theta, x_i, y_i), y_i)\}_{i=0}^N$ *where $(x_i, y_i) \sim \widehat{\mathcal{D}}_n$ consists of $N$ i.i.d. samples from $\mathcal{D}_a$.*

We define the population loss as $\mathcal{L}_\mathcal{D}(\theta) := \mathbb{E}_{(x,y)\sim\mathcal{D}}\mathcal{L}(f(x), y)$, and let $g_\mathcal{D}(\theta) := \nabla\mathcal{L}_\mathcal{D}(\theta)$. For simplification, we use $g_a := \nabla\mathcal{L}_{\mathcal{D}_a}(\theta)$, $\widehat{g}_a := \nabla\mathcal{L}_{\widehat{\mathcal{D}}_a}(\theta)$, and $\widehat{g}_n := \nabla\mathcal{L}_{\widehat{\mathcal{D}}_n}(\theta)$. $g_{GP} = \beta \cdot g_p + (1 - \beta) \cdot \widehat{g}_a$ (Definition A.3) is the aggregation using GP. We define the following optimization problem.

**Definition 4.2** (Aggregation for NT and AT). *$f_\theta$ is trained by iteratively updating the parameter*

$$\theta \leftarrow \theta - \mu \cdot \texttt{Aggr}(\widehat{g}_a, \widehat{g}_n),$$

*where $\mu$ is the step size. We seek an aggregation rule $\texttt{Aggr}(\cdot) = \widehat{g}_{\texttt{Aggr}}$ such that after training, $f_\theta$ minimizes the population loss function $\mathcal{L}_{\mathcal{D}_a}(\theta)$.*

We need $\widehat{g}_{\texttt{Aggr}}$ to be close to $g_a$ for each iteration, since $g_a$ is the optimal update on $\mathcal{D}_a$. Thus, we define $L^\pi$-Norm and delta error to indicate the performance of different aggregation rules.

**Definition 4.3** ($L^\pi$-Norm [Enyi Jiang, 2024]). *Given a distribution $\pi$ on the parameter space $\theta$, we define an inner product $\langle g_\mathcal{D}, g_{\mathcal{D}'} \rangle_\pi = \mathbb{E}_{\theta\sim\pi}[\langle g_\mathcal{D}(\theta), g_{\mathcal{D}'}(\theta) \rangle]$. The inner product induces the $L^\pi$-norm on $g_\mathcal{D}$ as $\|g_\mathcal{D}\|_\pi := \sqrt{\mathbb{E}_{\theta\sim\pi}\|g_\mathcal{D}(\theta)\|^2}$. We use $L^\pi$-norm to measure the gradient differences under certain $\mathcal{D}$.*

**Definition 4.4** (Delta Error of an aggregation rule $\texttt{Aggr}(\cdot)$). *We define the following squared error term to measure the closeness between $\widehat{g}_{\texttt{Aggr}}$ and $g_a$ under $\widehat{\mathcal{D}}_a^t$ (distribution at time step $t$), i.e.,*

$$\Delta^2_{\texttt{Aggr}} := \mathbb{E}_{\widehat{\mathcal{D}}_a^t}\|g_a - \widehat{g}_{\texttt{Aggr}}\|_\pi^2.$$

Delta errors $\Delta^2_{AT}$ and $\Delta^2_{GP}$ measure the closesness of $g_{GP}, \widehat{g}_a$ from $g_a$ in $\widehat{\mathcal{D}}_a$ at each iteration.

**Theorem 4.5** (Error Analysis of GP). *When the model dimension $m \to \infty$, for an epoch $t$, we have an approximation of the error difference $\Delta^2_{AT} - \Delta^2_{GP}$ as follows*

$$\Delta^2_{AT} - \Delta^2_{GP} \approx \beta(2 - \beta)\mathbb{E}_{\widehat{\mathcal{D}}_a^t}\|g_a - \widehat{g}_a\|_\pi^2 - \beta^2\bar{\tau}^2\|g_a - \widehat{g}_n\|_\pi^2$$

*$\bar{\tau}^2 = \mathbb{E}_\pi[\tau^2] \in [0, 1]$, where $\tau(\theta)$ is the $\sin(\cdot)$ value of the angle between $\widehat{g}_n$ and $g_a - \widehat{g}_n$.*

Theorem 4.5 shows $\Delta^2_{GP}$ is generally smaller than $\Delta^2_{AT}$ for a large model dimension during each iteration, as is the case for the models in our evaluation, with $\beta = 0.5$, since $\beta(1 - \beta) > \beta^2(0.75 > 0.25)$ and the small value of $\bar{\tau}$ in practice (see Interpretation of Theorem A.2 in Appendix A, where we show the order of difference is between $1e^{-8}$ and $1e^{-12}$). Thus, GP achieves better robust accuracy than AT by achieving a smaller delta error; GP also obtains good clean accuracy by combining parts of the model updates from the clean distribution $\widehat{\mathcal{D}}_n$. Further, we provide an error analysis of a single gradient step in Theorem A.1 and convergence analysis in Theorem A.2, showing that a smaller Delta error results in better convergence. The full proof of all theorems is in Appendix A.

We outline the **AT-GP** method in Algorithm 2 and it can be extended to the multiple-norm scenario. The overhead of this algorithm comes from natural training and GP operation. Their costs are small, and we discuss this more in Section 5.2. Combining logit pairing and gradient projection methods, we provide the **RAMP** framework which is similar to Algorithm 2, except that we replace line 4 of Algorithm 2 as Algorithm 1 line 3-9.

## 5 Experiment

**Datasets, baselines, and models.** CIFAR-10 [Krizhevsky et al., 2009] includes 60K images with 50K and 10K images for training and testing respectively. ImageNet has $\approx 14.2$M images and 1K classes, containing $\approx 1.3$M training, 50K validation, and 100K test images [Russakovsky et al.,

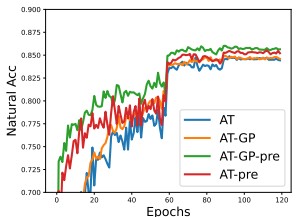

(a) Clean Accuracy

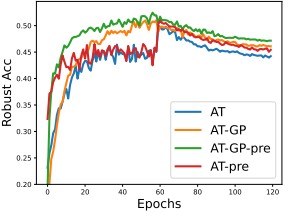

(b) Robust Accuracy: PGD-20

Figure 2: $l_\infty$ **AT-GP** with PGD [Madry et al., 2017] with $\epsilon = 0.031$ on CIFAR-10 improves accuracy and robustness. Pre-training on $\widehat{\mathcal{D}}_n$ for 50 epochs further boosts the performance.

|  | Clean | $l_\infty$ |
|---|---|---|
| RN-18 $l_\infty$ | 84.2 | 47.4 |
| RN-18 $l_\infty$-GP | 84.5 | **48.3** |
| RN-18 $l_\infty$-GP-pre | **84.9** | **48.3** |

Figure 3: $l_\infty$ **AT-GP** with APGD [Croce and Hein, 2020] improves robustness against $l_\infty$ AutoAttack [Croce and Hein, 2020] with $\epsilon = \frac{8}{255}$. RN-18 $l_\infty$-GP uses **AT-GP**; RN-18 $l_\infty$-GP-pre pre-trains 40 epochs on $\widehat{\mathcal{D}}_n$ before **AT-GP** is applied.

2015]. We compare **RAMP** with following baselines: 1. **SAT** [Madaan et al., 2021]: randomly sample one of the $l_1, l_2, l_\infty$ attacks. 2. **AVG** [Tramer and Boneh, 2019]: take the average of $l_1, l_2, l_\infty$ examples. 3. **MAX** [Tramer and Boneh, 2019]: take the worst of $l_1, l_2, l_\infty$ attacks. 4. **MSD** [Maini et al., 2020]: find the worst-case examples over $l_1, l_2, l_\infty$ steepest descent directions during each step of inner maximization. 5. **E-AT** [Croce and Hein, 2022]: randomly sample between $l_1, l_\infty$ attacks. For models, we use PreAct-ResNet-18, ResNet-50, WideResNet-34-20, and WideResNet-70-16 for CIFAR-10, as well as ResNet-50 and XCiT-S transformer for ImageNet.

**Implementations and Evaluation.** For AT from scratch for CIFAR-10, we train PreAct ResNet-18 [He et al., 2016] with a $lr = 0.05$ for 70 epochs and 0.005 for 10 more epochs. We set $\lambda = 2$, $\beta = 0.5$ for training from scratch, and $\lambda = 0.5$ for robust fine-tuning. For all methods, we use 10 steps for the inner maximization in AT. For ImageNet, we perform 1 epoch of fine-tuning and use a learning rate $lr = 0.005$, $\lambda = 0.5$ for ResNet-50 and $lr = 1e^{-4}$, $\lambda = 0.5$ for XCiT-S models. We reduce the rate by a factor of 10 every $\frac{1}{3}$ of the training epoch and set the weight decay to $1e^{-4}$. We use APGD with 5 steps for $l_\infty$ and $l_2$, 15 steps for $l_1$. Settings are similar to [Croce and Hein, 2022]. We use the standard values of $\epsilon_1 = 12, \epsilon_2 = 0.5, \epsilon_\infty = \frac{8}{255}$ for CIFAR-10 and $\epsilon_1 = 255, \epsilon_2 = 2, \epsilon_\infty = \frac{4}{255}$ for ImageNet. We focus on $l_\infty$-AT models for fine-tuning, as Croce and Hein [2022] shows their higher union accuracy for the $\epsilon$ values in our evaluation. We report the clean accuracy, robust accuracy against $\{l_1, l_2, l_\infty\}$ attacks, union accuracy, universal robustness against common corruptions and unseen adversaries, as well as runtime for **RAMP**. The robust accuracy is evaluated using Autoattack [Croce and Hein, 2020]. More implementation details are in Appendix B.

## 5.1 Main Results

Table 1: **Different epsilon values**: **RAMP** consistently outperforms E-AT and MAX for both training from scratch and robust fine-tuning when the key tradeoff pair changes.

|  |  | $(12, 0.5, \frac{2}{255})$ |  |  |  |  | $(12, 1.5, \frac{8}{255})$ |  |  |  |  |
|---|---|---|---|---|---|---|---|---|---|---|---|
|  |  | Clean | $l_\infty$ | $l_2$ | $l_1$ | Union | Clean | $l_\infty$ | $l_2$ | $l_1$ | Union |
| Training from Scratch | E-AT | 87.2 | 73.3 | 64.1 | 55.4 | 55.4 | 83.5 | 41.0 | 25.5 | 52.9 | 25.5 |
|  | MAX | 85.6 | 72.1 | 63.6 | 56.4 | 56.4 | 74.6 | 42.9 | 35.7 | 50.3 | 35.6 |
|  | **RAMP** | 86.3 | 73.3 | 64.9 | 59.1 | **59.1** | 74.4 | 43.4 | 37.2 | 51.1 | **37.1** |
| Robust Fine-tuning | E-AT | 86.5 | 74.8 | 66.7 | 57.9 | 57.9 | 80.2 | 42.8 | 31.5 | 52.4 | 31.5 |
|  | MAX | 85.7 | 74.0 | 66.2 | 60.0 | 60.0 | 74.8 | 43.8 | 36.7 | 50.2 | 36.6 |
|  | **RAMP** | 85.8 | 74.0 | 66.2 | 60.1 | **60.1** | 74.9 | 43.7 | 37.0 | 50.2 | **36.9** |

**Robust fine-tuning.** In Table 2, we apply **RAMP** to larger models and datasets (ImageNet). However, the implementation of other baselines is not publicly available and Croce and Hein [2022] do not report other baseline results except E-AT on larger models and datasets, so we only compare against E-AT in Table 2, which shows **RAMP** consistently obtains better union accuracy and accuracy-robustness tradeoff than E-AT. We observe that **RAMP** improves the performance more as the model becomes larger. We obtain the SOTA union accuracy of 53.3% on CIFAR-10 and 29.1% on ImageNet.

**RAMP with varying $\epsilon_1, \epsilon_2, \epsilon_\infty$ values.** We provide results with 1. $(\epsilon_1 = 12, \epsilon_2 = 0.5, \epsilon_\infty = \frac{2}{255})$ where $\epsilon_\infty$ size is small and 2. $(\epsilon_1 = 12, \epsilon_2 = 1.5, \epsilon_\infty = \frac{8}{255})$ where $\epsilon_2$ size is large, using PreAct ResNet-18 model for CIFAR-10 dataset: these cases have different tradeoff pair compared to

Table 2: **Robust fine-tuning on larger models and datasets** (* uses extra data for pre-training). We evaluate all CIFAR-10 and Imagenet test points. **RAMP** consistently achieves better union accuracy with significant margins and good accuracy-robustness tradeoff.

| | Models | Methods | Clean | $l_\infty$ | $l_2$ | $l_1$ | Union |
|---|---|---|---|---|---|---|---|
| | WRN-70-16-$l_\infty$ (*) [Gowal et al., 2020] | E-AT | 89.6 | 54.4 | 76.7 | 58.0 | 51.6 |
| | | **RAMP** | 90.6 | 54.7 | 74.6 | 57.9 | **53.3** |
| | WRN-34-20-$l_\infty$ [Gowal et al., 2020] | E-AT | 87.8 | 49.0 | 71.6 | 49.8 | 45.1 |
| | | **RAMP** | 87.1 | 49.7 | 70.8 | 50.4 | **46.9** |
| **CIFAR-10** | WRN-28-10-$l_\infty$ (*) [Carmon et al., 2019] | E-AT | 89.3 | 51.8 | 74.6 | 53.3 | 47.9 |
| | | **RAMP** | 89.2 | 55.9 | 74.7 | 55.7 | **52.7** |
| | WRN-28-10-$l_\infty$ (*) [Gowal et al., 2020] | E-AT | 89.8 | 54.4 | 76.1 | 56.0 | 50.5 |
| | | **RAMP** | 89.4 | 55.9 | 74.7 | 56.0 | **52.9** |
| | RN-50-$l_\infty$ [Engstrom et al., 2019] | E-AT | 85.3 | 46.5 | 68.3 | 45.3 | 41.6 |
| | | **RAMP** | 84.3 | 47.0 | 67.7 | 46.5 | **43.3** |
| **ImageNet** | XCiT-S-$l_\infty$ [Debenedetti and Troncoso—EPFL, 2022] | E-AT | 68.4 | 38.1 | 51.8 | 23.8 | 23.4 |
| | | **RAMP** | 66.0 | 35.7 | 50.2 | 30.0 | **29.1** |
| | RN-50-$l_\infty$ [Engstrom et al., 2019] | E-AT | 58.2 | 26.9 | 39.5 | 18.8 | 17.8 |
| | | **RAMP** | 55.6 | 25.1 | 38.3 | 22.4 | **20.9** |

Figure 1. The pair identified using our heuristic are $l_1$ - $l_2$ and $l_2$ - $l_\infty$. In Table 1, we observe that **RAMP** consistently outperforms E-AT and MAX with significant margins in union accuracy, when training from scratch and performing robust fine-tuning. In Table 1, when $l_2$ is the bottleneck, E-AT obtains a lower union accuracy as it does not leverage $l_2$ examples. Similar observations are made across various epsilon values, with **RAMP** consistently outperforming other baselines, as detailed in Appendix B.4. Appendix B includes more training details/results, and ablation studies. Results for applying the trades loss to **RAMP** outperforming E-AT are detailed in Appendix B.6. Appendix B.7 presents robust fine-tuning using ResNet-18, where **RAMP** achieves the highest union accuracy.

**Adversarial training from random initialization.** Table 3 presents the results of AT from random initialization on CIFAR-10 with PreAct ResNet-18. **RAMP** has the highest union accuracy with good clean accuracy, which indicates that **RAMP** can mitigate the tradeoffs among perturbations and robustness/accuracy in this setting. The results for all baselines are from Croce and Hein [2022].

Table 3: **RN-18 model trained from random initialization** on CIFAR-10 over 5 trials: **RAMP** achieves the best union robustness and good clean accuracy compared with other baselines. Baseline results are from Croce and Hein [2022].

| Methods | Clean | $l_\infty$ | $l_2$ | $l_1$ | Union |
|---|---|---|---|---|---|
| SAT | 83.9±0.8 | 40.7±0.7 | 68.0±0.4 | 54.0±1.2 | 40.4±0.7 |
| AVG | 84.6±0.3 | 40.8±0.7 | 68.4±0.7 | 52.1±0.4 | 40.1±0.8 |
| MAX | 80.4±0.5 | 45.7±0.9 | 66.0±0.4 | 48.6±0.8 | 44.0±0.7 |
| MSD | 81.1±1.1 | 44.9±0.6 | 65.9±0.6 | 49.5±1.2 | 43.9±0.8 |
| E-AT | 82.2±1.8 | 42.7±0.7 | 67.5±0.5 | 53.6±0.1 | 42.4±0.6 |
| **RAMP** ($\lambda$=5) | 81.2±0.3 | 46.0±0.5 | 65.8±0.2 | 48.3±0.6 | **44.6±0.6** |
| **RAMP** ($\lambda$=2) | 82.1±0.3 | 45.5±0.3 | 66.6±0.3 | 48.4±0.2 | 44.0±0.2 |

Table 4: Individual, average, and union accuracy against common corruptions (averaged across five levels) and unseen adversaries using WideResNet-28-10 on CIFAR-10 dataset.

| Models | Common Corruptions | $l_0$ | fog | snow | gabor | elastic | jpeginf | Avg | Union |
|---|---|---|---|---|---|---|---|---|---|
| $l_1$-AT | 78.2 | 79.0 | 41.4 | 22.9 | 40.5 | 48.9 | 48.4 | 46.9 | 12.8 |
| $l_2$-AT | 77.2 | 67.5 | 48.7 | 26.1 | 44.1 | 53.2 | 45.4 | 47.5 | 16.2 |
| $l_\infty$-AT | 73.4 | 55.5 | 44.7 | 32.9 | 53.8 | 56.6 | 33.4 | 46.2 | 19.1 |
| Winninghand [Diffenderfer et al., 2021] | **91.1** | 74.1 | 74.5 | 18.3 | 76.5 | 12.6 | 0.0 | 42.7 | 0.0 |
| E-AT | 71.5 | 58.5 | 35.9 | 35.3 | 50.7 | 55.7 | 60.3 | 49.4 | 21.9 |
| MAX | 71.0 | 56.2 | 42.9 | 35.4 | 49.8 | 57.8 | 55.7 | 49.6 | 24.4 |
| **RAMP** | 75.5 | 55.5 | 40.5 | 40.2 | 52.9 | 60.3 | 56.1 | **50.9** | **26.1** |

**Universal Robustness.** In Table 4, we report average accuracy against common corruptions and union accuracy against unseen adversaries from Laidlaw et al. [2020] (implementation details are in Appendix B.3). We compare against $l_p$ pretrained models, E-AT, MAX, winninghand [Diffenderfer et al., 2021] (a SOTA method for natural corruptions) using WideResNet-28-10 architecture on the CIFAR-10 dataset. Compared to E-AT and MAX, **RAMP** achieves 4% higher accuracy for common corruptions with five severity levels and 2-4% better union accuracy against multiple unseen adversaries. Winninghand has high corruption robustness but no adversarial robustness. The results show that **RAMP** obtains a better robustness and accuracy tradeoff with stronger universal robustness. In Appendix B.3, we evaluate on ResNet-18 to support this fact further.

## 5.2 Ablation Study and Discussion

**Sensitivities of $\lambda$.** We perform experiments with different $\lambda$ values in $[0.1, 0.5, 1.0, 1.5, 2, 3, 4, 5]$ for robust fine-tuning and $[1.5, 2, 3, 4, 5, 6]$ for AT from scratch using PreAct-ResNet-18 model for CIFAR-10 dataset. In Figure 4, we observe a decreased clean accuracy when $\lambda$ becomes larger. We

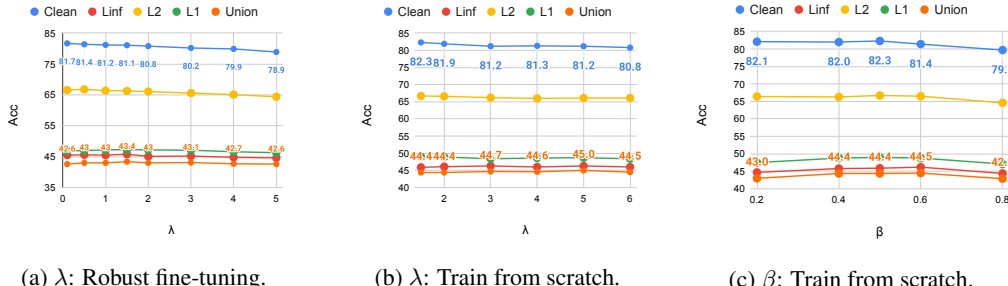

(a) $\lambda$: Robust fine-tuning.  (b) $\lambda$: Train from scratch.  (c) $\beta$: Train from scratch.

Figure 4: Alabtion studies on $\lambda$ and $\beta$ hyper-parameters.

pick $\lambda = 2.0$ for training from scratch (Figure 4a) and $\lambda = 0.5$ for robust fine-tuning (Figure 4b) in our main experiments, as these values of $\lambda$ yield both good clean and union accuracy.

**Choices of $\beta$.** Figure 4c shows the performance of **RAMP** with varying $\beta$ values on CIFAR-10 ResNet-18 experiments. We pick $\beta = 0.5$ for combining natural training and AT via GP, which achieves comparatively good robustness and clean accuracy. This choice is also based on Theorem 4.5 when $\beta(2 - \beta)$ has the largest difference from $\beta^2$ (0.75 vs 0.25).

**Fine-tune $l_p$ AT models with RAMP.** Table 5 shows the robust fine-tuning results using **RAMP** with $l_\infty$-AT ($q = \infty, r = 1$), $l_1$-AT ($q = 1, r = \infty$), $l_2$-AT ($q = \infty, r = 1$) RN-18 models for CIFAR-10 dataset. For $l_\infty - l_1$ tradeoffs, RAMP on $l_\infty$-AT pre-trained model achieves the best union accuracy.

|  | Clean | $l_\infty$ | $l_2$ | $l_1$ | Union |
|---|---|---|---|---|---|
| RN-18 $l_\infty$-AT | 81.5 | **45.5** | 66.4 | **47.0** | **42.9** |
| RN-18 $l_1$-AT | 81.0 | 42.6 | 66.0 | 48.1 | 41.5 |
| RN-18 $l_2$-AT | **84.1** | 41.6 | **69.1** | 45.4 | 39.4 |

Table 5: **RAMP** with $l_\infty, l_1, l_2$-RN-18-AT models on CIFAR-10 with standard epsilons.

**Computational analysis and Limitations.** The extra training costs of AT-GP are small, e.g. for each epoch on ResNet-18, the extra NT takes 6 seconds and the standard AT takes 78 seconds using a single NVIDIA A100 GPU, and the **GP** operation only takes 0.04 seconds on average. RAMP is more expensive than E-AT and less expensive than MAX. We have a complete runtime analysis in Appendix B.2. We notice occasional drops in clean accuracy during fine-tuning with **RAMP**. In some cases, union accuracy improves slightly but clean accuracy and single $l_p$ robustness reduce. Further, we find no negative societal impact from this work.

## 6  Conclusion

We introduce **RAMP**, a framework enhancing multiple-norm robustness and achieving superior *universal robustness* against corruptions and perturbations by addressing tradeoffs among $l_p$ perturbations and accuracy/robustness. We apply a new logit pairing loss and use gradient projection to obtain SOTA union accuracy with favorable accuracy/robustness tradeoffs against common corruptions and other unseen adversaries. Results demonstrate that **RAMP** surpasses SOTA methods in union accuracy across model architectures on CIFAR-10 and ImageNet.

## Acknowledgments

This work was supported in part by NSF Grants No. CCF-2238079, CCF-2316233, CNS-2148583. We would like to thank Jacky Yibo Zhang for the helpful discussions and advice on the proof. Also, we thank anonymous reviewers for their valuable feedback on the paper.

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

# A Proof of Theorems

## A.1 Proof of Theorem A.2

We first show what happens during one step of optimization, where we highlight the importance of analyzing delta error.

**Theorem A.1.** *Consider model parameter $\theta \sim \pi$ and an aggregation rule $\texttt{Aggr}(\cdot)$ with step size $\mu > 0$. Define the updated parameter as*

$$\theta^+ := \theta - \mu \widehat{g}_{Aggr}(\theta).$$

*Assuming the gradient $\nabla \mathcal{L}(\theta)$ is $\gamma$-Lipschitz in $\theta$ for any input, and let the step size $\mu \leq \frac{1}{\gamma}$, we have*

$$\mathbb{E}_{\widehat{\mathcal{D}}_a, \theta}[\mathcal{L}_{\mathcal{D}_a}(\theta^+) - \mathcal{L}_{\mathcal{D}_a}(\theta)] \leq -\tfrac{\mu}{2}(\|g_a\|_\pi^2 - \Delta_{Aggr}^2).$$

*Proof.* The proof is the same as Theorem A.1 in [Enyi Jiang, 2024]. □

**Theorem A.2** (Convergence of $\texttt{Aggr}(\cdot)$)**.** *For any probability measure $\pi$ over the parameter space, and an aggregation rule $\texttt{Aggr}(\cdot)$ with step size $\mu > 0$. We update the parameter for $T$ steps by $\theta^{t+1} := \theta^t - \mu \widehat{g}_{Aggr}(\theta^t)$. Assume the gradient $\nabla \mathcal{L}(\theta)$ and $\widehat{g}_{Aggr}(\theta)$ are $\frac{\gamma}{2}$-Lipschitz in $\theta$ such that $\theta^t \to \widehat{\theta}_{Aggr}$. $\Delta_{Aggr\_max}$ is the Delta error at time $t'$ when $\|\widehat{g}_{Aggr}(\widehat{\theta}_{Aggr}) - \nabla \mathcal{L}_{\mathcal{D}_a^{t'}}(\widehat{\theta}_{Aggr})\|^2$ is maximized. Then, given step size $\mu \leq \frac{1}{\gamma}$ and a small enough $\epsilon > 0$, with probability at least $1 - \delta$ we have*

$$\|\nabla \mathcal{L}_{\mathcal{D}_a^T}(\theta^T)\|^2 \leq \frac{1}{\delta^2}\left(\sqrt{C_\epsilon \cdot \Delta_{Aggr\_max}^2} + \mathcal{O}(\epsilon)\right)^2 + \mathcal{O}\left(\frac{1}{T}\right) + \mathcal{O}(\epsilon),$$

*where $C_\epsilon = \mathbb{E}_{\widehat{\mathcal{D}}_a^{t'}}[1/\pi(B_\epsilon(\widehat{\theta}_{Aggr}))]^2$ and $B_\epsilon(\widehat{\theta}_{Aggr}) \subset \mathbb{R}^m$ is the ball with radius $\epsilon$ centered at $\widehat{\theta}_{Aggr}$. The $C_\epsilon$ measures how well $\pi$ covers where the optimization goes.*

*Proof.* Denote random function $\widehat{f} : \mathbb{R}^m \to \mathbb{R}_+$ as

$$\widehat{f}(\theta) = \|\widehat{g}_{\texttt{Aggr}}(\theta) - \nabla \mathcal{L}_{\mathcal{D}_a}(\theta)\|, \tag{8}$$

where the randomness comes from $\widehat{\mathcal{D}}_a$. Note that $\widehat{f}$ is $\gamma$-Lipschitz by assumption. Now we consider $B_\epsilon(\widehat{\theta}_{\texttt{Aggr}}) \subset \mathbb{R}^m$, i.e., the ball with radius $\epsilon$ centered at $\widehat{\theta}_{\texttt{Aggr}}$. Then, by $\gamma$-Lipschitzness we have

$$\mathbb{E}_{\theta \sim \pi} \widehat{f}(\theta) = \int \widehat{f}(\theta)\,\mathrm{d}\pi(\theta)$$

$$\geq \int_{B_\epsilon(\widehat{\theta}_{\texttt{Aggr}})} (\widehat{f}(\widehat{\theta}_{\texttt{Aggr}}) - \gamma\epsilon)\,\mathrm{d}\pi(\theta)$$

$$= (\widehat{f}(\widehat{\theta}_{\texttt{Aggr}}) - \gamma\epsilon)\pi(B_\epsilon(\widehat{\theta}_{\texttt{Aggr}}))$$

Therefore,

$$\widehat{f}(\widehat{\theta}_{\texttt{Aggr}}) \leq \frac{1}{\pi(B_\epsilon(\widehat{\theta}_{\texttt{Aggr}}))} \cdot \mathbb{E}_{\theta \sim \pi} \widehat{f}(\theta) + \mathcal{O}(\epsilon).$$

Taking expectation w.r.t. $\widehat{\mathcal{D}}_a$ on both sides, we have

$$\mathbb{E}_{\widehat{\mathcal{D}}_a} \widehat{f}(\widehat{\theta}_{\texttt{Aggr}}) \leq \mathbb{E}_{\widehat{\mathcal{D}}_a}\left[\frac{1}{\pi(B_\epsilon(\widehat{\theta}_{\texttt{Aggr}}))} \cdot \mathbb{E}_{\theta \sim \pi} \widehat{f}(\theta)\right] + \mathcal{O}(\epsilon)$$

$$\leq \sqrt{\mathbb{E}_{\widehat{\mathcal{D}}_a}\left[\frac{1}{\pi(B_\epsilon(\widehat{\theta}_{\texttt{Aggr}}))}\right]^2 \cdot \mathbb{E}_{\widehat{\mathcal{D}}_a}\left[\mathbb{E}_{\theta \sim \pi} \widehat{f}(\theta)\right]^2} + \mathcal{O}(\epsilon) \qquad \text{(Cauchy-Schwarz)}$$

$$= \sqrt{C_\epsilon \cdot \mathbb{E}_{\widehat{\mathcal{D}}_a}\left[\mathbb{E}_{\theta \sim \pi} \widehat{f}(\theta)\right]^2} + \mathcal{O}(\epsilon) \qquad \text{(by definition of } C_\epsilon\text{)}$$

$$\leq \sqrt{C_\epsilon \cdot \mathbb{E}_{\widehat{\mathcal{D}}_a} \mathbb{E}_{\theta \sim \pi}\left[\widehat{f}(\theta)\right]^2} + \mathcal{O}(\epsilon) \qquad \text{(Jensen's inequality)}$$

$$= \sqrt{C_\epsilon \cdot \Delta_{\texttt{Aggr}}^2} + \mathcal{O}(\epsilon)$$

By Markov's inequality, with probability at least $1 - \delta$ we have a sampled dataset $\widehat{\mathcal{D}}_a$ such that

$$\widehat{f}(\widehat{\theta}_{\text{Aggr}}) \leq \frac{1}{\delta} \mathbb{E}_{\widehat{\mathcal{D}}_a} \widehat{f}(\widehat{\theta}_{\text{Aggr}}) \leq \frac{1}{\delta} \sqrt{C_\epsilon \cdot \Delta_{\text{Aggr}}^2} + \mathcal{O}(\epsilon/\delta) \tag{9}$$

Conditioned on such event, we proceed on to the optimization part.

Note that Theorem A.1 characterizes how the optimization works for one gradient update. We denote $\mathcal{D}_a^t$ as the data distribution $\mathcal{D}_a$ at time step $t$. Therefore, for any time step $t = 0, \ldots, T-1$, we can apply Theorem A.1 which only requires the Lipschitz assumption:

$$\mathcal{L}_{\mathcal{D}_a^t}(\theta^{t+1}) - \mathcal{L}_{\mathcal{D}_a^t}(\theta^t) \leq -\frac{\mu}{2} \left( \|\nabla \mathcal{L}_{\mathcal{D}_a^t}(\theta^t)\|^2 - \|\widehat{g}_{\text{Aggr}}(\theta^t) - \nabla \mathcal{L}_{\mathcal{D}_a^t}(\theta^t)\|^2 \right).$$

We notice that $\mathcal{D}_a^t$ changes based on $\theta$s of different time steps. On both sides, to sum over $t = 0, \ldots, T-1$, we first consider two terms:

$$(\mathcal{L}_{\mathcal{D}_a^t}(\theta^{t+1}) - \mathcal{L}_{\mathcal{D}_a^t}(\theta^t)) + (\mathcal{L}_{\mathcal{D}_a^{t-1}}(\theta^t) - \mathcal{L}_{\mathcal{D}_a^{t-1}}(\theta^{t-1}))$$

To compare $\mathcal{L}_{\mathcal{D}_a^t}(\theta^t)$ and $\mathcal{L}_{\mathcal{D}_a^{t-1}}(\theta^t)$, since $\mathcal{D}_a^t$ optimizes one more step than $\mathcal{D}_a^{t-1}$, we assume $\mathcal{L}_{\mathcal{D}_a^t}(\theta^t) \leq \mathcal{L}_{\mathcal{D}_a^{t-1}}(\theta^t)$ for $\forall t$. Therefore, we have:

$$(\mathcal{L}_{\mathcal{D}_a^t}(\theta^{t+1}) - \mathcal{L}_{\mathcal{D}_a^t}(\theta^t)) + (\mathcal{L}_{\mathcal{D}_a^{t-1}}(\theta^t) - \mathcal{L}_{\mathcal{D}_a^{t-1}}(\theta^{t-1})) \geq (\mathcal{L}_{\mathcal{D}_a^t}(\theta^{t+1}) - \mathcal{L}_{\mathcal{D}_a^{t-1}}(\theta^t)) + (\mathcal{L}_{\mathcal{D}_a^{t-1}}(\theta^t) - \mathcal{L}_{\mathcal{D}_a^{t-1}}(\theta^{t-1}))$$

Summing up all time steps,

$$\mathcal{L}_{\mathcal{D}_a^t}(\theta^{t+1}) - \mathcal{L}_{\mathcal{D}_a^0}(\theta^0) \leq \mathcal{L}_{\mathcal{D}_a^t}(\theta^{t+1}) - \mathcal{L}_{\mathcal{D}_a^{t-1}}(\theta^t) + \mathcal{L}_{\mathcal{D}_a^{t-1}}(\theta^t) - \mathcal{L}_{\mathcal{D}_a^{t-2}}(\theta^{t-1}) + \ldots - \mathcal{L}_{\mathcal{D}_a^0}(\theta^0)$$

$$\leq -\frac{\mu}{2} \left( \sum_{t=0}^{T-1} \|\nabla \mathcal{L}_{\mathcal{D}_a^{t-1}}(\theta^t)\|^2 - \sum_{t=0}^{T-1} \|\widehat{g}_{\text{Aggr}}(\theta^t) - \nabla \mathcal{L}_{\mathcal{D}_a^t}(\theta^t)\|^2 \right).$$

Dividing both sides by $T$, and with regular algebraic manipulation we derive

$$\frac{1}{T} \sum_{t=0}^{T-1} \|\nabla \mathcal{L}_{\mathcal{D}_a^{t-1}}(\theta^t)\|^2 \leq \frac{2}{\mu T} (\mathcal{L}_{\mathcal{D}_a^0}(\theta^0) - \mathcal{L}_{\mathcal{D}_a^{T-1}}(\theta^T)) + \frac{1}{T} \sum_{t=0}^{T-1} \|\widehat{g}_{\text{Aggr}}(\theta^t) - \nabla \mathcal{L}_{\mathcal{D}_a^t}(\theta^t)\|^2.$$

Note that we assume the loss function $\mathcal{L}_{\mathcal{D}}(\theta) := \mathbb{E}_{(x,y) \sim \mathcal{D}} \mathcal{L}(f(x), y)$, is non-negative. Thus, we have

$$\frac{1}{T} \sum_{t=0}^{T-1} \|\nabla \mathcal{L}_{\mathcal{D}_a^{t-1}}(\theta^t)\|^2 \leq \frac{2\mathcal{L}_{\mathcal{D}_a^0}(\theta^0)}{\mu T} + \frac{1}{T} \sum_{t=0}^{T-1} \|\widehat{g}_{\text{Aggr}}(\theta^t) - \nabla \mathcal{L}_{\mathcal{D}_a^t}(\theta^t)\|^2. \tag{10}$$

Note that we assume given $\widehat{\mathcal{D}}_a$ we have $\theta^t \to \widehat{\theta}_{\text{Aggr}}$. Therefore, for any $\epsilon > 0$ there exist $T_\epsilon$ such that

$$\forall t > T_\epsilon : \|\theta^t - \widehat{\theta}_{\text{Aggr}}\| < \epsilon. \tag{11}$$

This implies that $\forall t > T_\epsilon$:

$$\mu \|\widehat{g}_{\text{Aggr}}(\theta^t)\| = \|\theta^{t+1} - \widehat{\theta}_{\text{Aggr}} + \widehat{\theta}_{\text{Aggr}} - \theta^t\| \leq \|\theta^{t+1} - \widehat{\theta}_{\text{Aggr}}\| + \|\widehat{\theta}_{\text{Aggr}} - \theta^t\| < 2\epsilon. \tag{12}$$

Moreover, (11) also implies $\forall t_1, t_2 > T_\epsilon$:

$$\|\nabla \mathcal{L}_{\mathcal{D}_a^{t_1}}(\theta^{t_1}) - \nabla \mathcal{L}_{\mathcal{D}_a^{t_2}}(\theta^{t_2})\| \leq \gamma \|\theta^{t_1} - \theta^{t_2}\| \qquad (\gamma\text{-Lipschitzness})$$

$$< 2\epsilon. \tag{13}$$

Now, let's get back to (10). For $\forall T > T_\epsilon$ we have

$$\frac{1}{T}\sum_{t=0}^{T-1}\|\nabla\mathcal{L}_{\mathcal{D}_a^{t-1}}(\theta^t)\|^2 \le \frac{2\mathcal{L}_{\mathcal{D}_a^0}(\theta^0)}{\mu T} + \frac{1}{T}\sum_{t=0}^{T_\epsilon-1}\|\widehat{g}_{\text{Aggr}}(\theta^t) - \nabla\mathcal{L}_{\mathcal{D}_a^t}(\theta^t)\|^2 + \frac{1}{T}\sum_{t=T_\epsilon}^{T-1}\|\widehat{g}_{\text{Aggr}}(\theta^t) - \nabla\mathcal{L}_{\mathcal{D}_a^t}(\theta^t)\|^2$$

$$= \mathcal{O}\left(\frac{1}{T}\right) + \frac{1}{T}\sum_{t=T_\epsilon}^{T-1}\|\widehat{g}_{\text{Aggr}}(\theta^t) - \nabla\mathcal{L}_{\mathcal{D}_a^t}(\theta^t)\|^2$$

$$= \mathcal{O}\left(\frac{1}{T}\right) + \frac{1}{T}\sum_{t=T_\epsilon}^{T-1}\|\widehat{g}_{\text{Aggr}}(\theta^t) - \widehat{g}_{\text{Aggr}}(\widehat{\theta}_{\text{Aggr}}) + \widehat{g}_{\text{Aggr}}(\widehat{\theta}_{\text{Aggr}}) - \nabla\mathcal{L}_{\mathcal{D}_a^t}(\theta^t)\|^2$$

$$\le \mathcal{O}\left(\frac{1}{T}\right) + \frac{1}{T}\sum_{t=T_\epsilon}^{T-1}\left(\|\widehat{g}_{\text{Aggr}}(\theta^t) - \widehat{g}_{\text{Aggr}}(\widehat{\theta}_{\text{Aggr}})\| + \|\widehat{g}_{\text{Aggr}}(\widehat{\theta}_{\text{Aggr}}) - \nabla\mathcal{L}_{\mathcal{D}_a^t}(\theta^t)\|\right)^2$$

(triangle inequality)

$$= \mathcal{O}\left(\frac{1}{T}\right) + \frac{1}{T}\sum_{t=T_\epsilon}^{T-1}\left(\mathcal{O}(\epsilon) + \|\widehat{g}_{\text{Aggr}}(\widehat{\theta}_{\text{Aggr}}) - \nabla\mathcal{L}_{\mathcal{D}_a^t}(\theta^t)\|\right)^2 \quad \text{(by (12))}$$

$$= \mathcal{O}\left(\frac{1}{T}\right) + \mathcal{O}(\epsilon) + \frac{1}{T}\sum_{t=T_\epsilon}^{T-1}\left(\|\widehat{g}_{\text{Aggr}}(\widehat{\theta}_{\text{Aggr}}) - \nabla\mathcal{L}_{\mathcal{D}_a^t}(\widehat{\theta}_{\text{Aggr}}) + \nabla\mathcal{L}_{\mathcal{D}_a^t}(\widehat{\theta}_{\text{Aggr}}) - \nabla\mathcal{L}_{\mathcal{D}_a^t}(\theta^t)\|\right)^2$$

$$\le \mathcal{O}\left(\frac{1}{T}\right) + \mathcal{O}(\epsilon) + \frac{1}{T}\sum_{t=T_\epsilon}^{T-1}\left(\|\widehat{g}_{\text{Aggr}}(\widehat{\theta}_{\text{Aggr}}) - \nabla\mathcal{L}_{\mathcal{D}_a^t}(\widehat{\theta}_{\text{Aggr}})\| + \mathcal{O}(\epsilon)\right)^2$$

(by (13))

$$\le \mathcal{O}\left(\frac{1}{T}\right) + \mathcal{O}(\epsilon) + \|\widehat{g}_{\text{Aggr}}(\widehat{\theta}_{\text{Aggr}}) - \nabla\mathcal{L}_{\mathcal{D}_a^{t'}}(\widehat{\theta}_{\text{Aggr}})\|^2 \qquad (14)$$

Equation 14 bounds the left hand side with the maximum $\|\widehat{g}_{\text{Aggr}}(\widehat{\theta}_{\text{Aggr}}) - \nabla\mathcal{L}_{\mathcal{D}_a^t}(\widehat{\theta}_{\text{Aggr}})\|^2$ one can get during the optimization steps. Here, we assume at time $t'$, the largest value is attained. We denote $\Delta_{\text{Aggr\_max}}^2$ as the delta error at time step $t'$.

Then, we can continue with what we have done at the beginning of the proof of this theorem:

$$(14) = \mathcal{O}\left(\frac{1}{T}\right) + \mathcal{O}(\epsilon) + f(\widehat{\theta}_{\text{Aggr}})^2 \qquad \text{(by (8))}$$

$$\le \mathcal{O}\left(\frac{1}{T}\right) + \mathcal{O}(\epsilon) + \left(\frac{1}{\delta}\sqrt{C_\epsilon \cdot \Delta_{\text{Aggr\_max}}^2} + \mathcal{O}(\epsilon/\delta)\right)^2 \qquad \text{(by (9))}$$

Therefore, combining the above we finally have: for $\forall T > T_\epsilon$ with probability at least $1 - \delta$,

$$\frac{1}{T}\sum_{t=0}^{T-1}\|\nabla\mathcal{L}_{\mathcal{D}_a^{t-1}}(\theta^t)\|^2 \le \mathcal{O}\left(\frac{1}{T}\right) + \mathcal{O}(\epsilon) + \frac{1}{\delta^2}\left(\sqrt{C_\epsilon \cdot \Delta_{\text{Aggr\_max}}^2} + \mathcal{O}(\epsilon)\right)^2 \qquad (15)$$

To complete the proof, let us investigate the left-hand side.

$$\frac{1}{T}\sum_{t=0}^{T-1}\|\nabla\mathcal{L}_{\mathcal{D}_a^{t-1}}(\theta^t)\|^2 = \frac{1}{T}\sum_{t=0}^{T_\epsilon-1}\|\nabla\mathcal{L}_{\mathcal{D}_a^t}(\theta^t)\|^2 + \frac{1}{T}\sum_{t=T_\epsilon}^{T-1}\|\nabla\mathcal{L}_{\mathcal{D}_a^t}(\theta^t)\|^2$$

$$= \mathcal{O}\left(\frac{1}{T}\right) + \frac{1}{T}\sum_{t=T_\epsilon}^{T-1}\|\nabla\mathcal{L}_{\mathcal{D}_a^t}(\theta^t)\|^2$$

$$\geq \mathcal{O}\left(\frac{1}{T}\right) + \frac{1}{T}\sum_{t=T_\epsilon}^{T-1}\left(\|\nabla\mathcal{L}_{\mathcal{D}_a^t}(\theta^t) - \nabla\mathcal{L}_{\mathcal{D}_a^T}(\theta^T)\| - \|\nabla\mathcal{L}_{\mathcal{D}_a^T}(\theta^T)\|\right)^2$$

(triangle inequality)

$$= \mathcal{O}\left(\frac{1}{T}\right) + \frac{1}{T}\sum_{t=T_\epsilon}^{T-1}\left(\mathcal{O}(\epsilon) + \|\nabla\mathcal{L}_{\mathcal{D}_a^T}(\theta^T)\|^2\right) \qquad \text{(by (13))}$$

$$= \mathcal{O}\left(\frac{1}{T}\right) + \mathcal{O}(\epsilon) + \|\nabla\mathcal{L}_{\mathcal{D}_a^T}(\theta^T)\|^2. \qquad (16)$$

Combining (15) and (16), we finally have

$$\|\nabla\mathcal{L}_{\mathcal{D}_a^T}(\theta^T)\|^2 \leq \mathcal{O}\left(\frac{1}{T}\right) + \mathcal{O}(\epsilon) + \frac{1}{\delta^2}\left(\sqrt{C_\epsilon \cdot \Delta^2_{\texttt{Aggr\_max}}} + \mathcal{O}(\epsilon)\right)^2,$$

which completes the proof. $\qquad\square$

## A.2   Proof of Theorem 4.5

To prove Theorem 4.5, we first use the following definitions and lemmas from [Enyi Jiang, 2024], to get the delta errors of Gradient Projection (GP) and standard adversarial training (AT):

**Definition A.3** (GP Aggregation). *Let $\beta \in [0, 1]$ be the weight that balances between $\widehat{g}_a$ and $\widehat{g}_n$. The GP aggregation operation is*

$$GP(\widehat{g}_a, \widehat{g}_n) = \left((1-\beta)\widehat{g}_a + \beta\texttt{Proj}_+(\widehat{g}_a|\widehat{g}_n)\right).$$

*where $\texttt{Proj}_+(\widehat{g}_a|\widehat{g}_n) = \max\{\langle\widehat{g}_a, \widehat{g}_n\rangle, 0\}\widehat{g}_n/\|\widehat{g}_n\|^2$ is the operation that projects $\widehat{g}_a$ to the positive direction of $\widehat{g}_n$.*

**Definition A.4** (AT Aggregation). *The AT aggregation operation is*

$$AT(\widehat{g}_a) = \widehat{g}_a.$$

*standard AT only leverages the gradient update on $\widehat{\mathcal{D}}_a$.*

**Lemma A.5** (Delta Error of GP). *Given distributions $\widehat{\mathcal{D}}_a$, $\mathcal{D}_a$ and $\widehat{\mathcal{D}}_n$, as well as the model updates $\widehat{g}_a, g_a, \widehat{g}_n$ on these distributions per epoch, we have $\Delta^2_{GP}$ as follows*

$$\Delta^2_{GP} \approx \left((1-\beta)^2 + \frac{2\beta - \beta^2}{m}\right)\mathbb{E}_{\widehat{\mathcal{D}}_a}\|g_a - \widehat{g}_a\|^2_\pi + \beta^2\bar{\tau}^2\|g_a - \widehat{g}_n\|^2_\pi,$$

*In the above equation, $m$ is the model dimension and $\bar{\tau}^2 = \mathbb{E}_\pi[\tau^2] \in [0, 1]$ where $\tau(\theta)$ is the $\sin(\cdot)$ value of the angle between $\widehat{g}_n$ and $g_a - \widehat{g}_n$. $\|\cdot\|_\pi$ is the $\pi$-norm over the model parameter space.*

*Proof.* The proof is the same as Theorem 4.4 in Enyi Jiang [2024]. $\qquad\square$

**Lemma A.6** (Delta Error of AT). *Given distributions $\widehat{\mathcal{D}}_a$, $\mathcal{D}_a$ and $\widehat{\mathcal{D}}_n$, as well as the model updates $\widehat{g}_a, g_a, \widehat{g}_n$ on these distributions per epoch, we have $\Delta^2_{AT}$ as follows*

$$\Delta^2_{AT} = \mathbb{E}_{\widehat{\mathcal{D}}_a}\|g_a - \widehat{g}_a\|^2_\pi,$$

*where $\|\cdot\|_\pi$ is the $\pi$-norm over the model parameter space.*

Then, we prove Theorem 4.5.

**Theorem A.7** (Error Analysis of GP). *When the model dimension is large ($m \to \infty$) at time step t, we have*

$$\Delta_{AT}^2 - \Delta_{GP}^2 \approx \beta(2-\beta)\mathbb{E}_{\widehat{\mathcal{D}}_a^t}\|g_a - \widehat{g_a}\|_\pi^2 - \beta^2\bar{\tau}^2\|g_a - \widehat{g_n}\|_\pi^2.$$

$\bar{\tau}^2 = \mathbb{E}_\pi[\tau^2] \in [0,1]$ *where $\tau$ is the $\sin(\cdot)$ value of the angle between $\widehat{g_n}$ and $g_a - \widehat{g_n}$, $\|\cdot\|_\pi$ is the $\pi$-norm over the model parameter space.*

*Proof.* $\Delta_{\text{AT}}^2 - \Delta_{\text{GP}}^2$

$\approx \mathbb{E}_{\widehat{\mathcal{D}}_a^t}\|g_a - \widehat{g_a}\|_\pi^2 - \left((1-\beta)^2 + \frac{2\beta - \beta^2}{m}\right)\mathbb{E}_{\widehat{\mathcal{D}}_a}\|g_a - \widehat{g_a}\|_\pi^2 - \beta^2\bar{\tau}^2\|g_a - \widehat{g_n}\|_\pi^2$

$= \left(1 - \left((1-\beta)^2 + \frac{2\beta - \beta^2}{m}\right)\right)\mathbb{E}_{\widehat{\mathcal{D}}_a^t}\|g_a - \widehat{g_a}\|_\pi^2 - \beta^2\bar{\tau}^2\|g_a - \widehat{g_n}\|_\pi^2$

$= (1 + \frac{1}{m})\beta(2-\beta)\mathbb{E}_{\widehat{\mathcal{D}}_a^t}\|g_a - \widehat{g_a}\|_\pi^2 - \beta^2\bar{\tau}^2\|g_a - \widehat{g_n}\|_\pi^2$

When $m \to \infty$, we have a simplified version of the error difference as follows

$$\Delta_{AT}^2 - \Delta_{GP}^2 \approx \beta(2-\beta)\mathbb{E}_{\widehat{\mathcal{D}}_a^t}\|g_a - \widehat{g_a}\|_\pi^2 - \beta^2\bar{\tau}^2\|g_a - \widehat{g_n}\|_\pi^2$$

$\square$

**Interpretation.** When $\beta = 0.5$, we can usually show $\Delta_{AT}^2 > \Delta_{GP}^2$, because $\beta(2-\beta) > \beta^2\bar{\tau}^2 (0.75 > 0.25)$ for the coefficients of two terms. We estimate the actual values of terms $E_{\widehat{D}_{a^t}}\|g_a - \widehat{g_a}\|_\pi^2$ (variance), $\|g_a - \widehat{g_n}\|_\pi^2$ (bias), and $\bar{\tau}$ using the estimation methods in Enyi Jiang [2024]. Table 6 displays the values of those terms as well as the error differences on ResNet18 experiments at epoch $5, 10, 15, 20, 60$. We plot the changing of these terms on the ResNet18 experiment in Figure 5. The order of difference is always positive and usually smaller than $1e^{-08}$ and approaches the order of $1e^{-12}$ in the end.

Table 6: Estimations the actual values of terms $E_{\widehat{D}_{a^t}}\|g_a - \widehat{g_a}\|_\pi^2$ (variance), $\|g_a - \widehat{g_n}\|_\pi^2$ (bias), $\bar{\tau}$, and $\Delta_{AT}^2 - \Delta_{GP}^2$ (error differences) across different epochs.

| Terms / epochs | 5 | 10 | 15 | 20 | 60 |
|---|---|---|---|---|---|
| $E_{\widehat{D}_{a^t}}\|g_a - \widehat{g_a}\|_\pi^2$ | 4.6017e-08 | 2.0448e-09 | 6.9623e-10 | 6.4329e-10 | 2.3849e-11 |
| $\|g_a - \widehat{g_n}\|_\pi^2$ | 0.0007 | 9.9098e-05 | 4.4932e-05 | 3.7930e-05 | 2.8391e-06 |
| $\bar{\tau}$ | 0.0071 | 0.0052 | 0.0036 | 0.0038 | 0.0030 |
| $\Delta_{AT}^2 - \Delta_{GP}^2$ | 2.5335e-08 | 8.5709e-10 | 3.7609e-10 | 3.4487e-10 | 1.1574e-11 |

# B  Additional Experiment Information

In this section, we provide more training details, additional experiment results on the universal robustness of **RAMP** to common corruptions and unseen adversaries, runtime analysis of RAMP, additional ablation studies on different logit pairing losses, and AT from random initialization results on CIFAR-10 using WideResNet-28-10.

## B.1  More Training Details

We set the batch size to $128$ for the experiments on ResNet-18 and WideResNet-28-10 architectures. We use an SGD optimizer with $0.9$ momentum and $5e^{-4}$ weight decay. For other experiments on ImageNet, we use a batch size of $64$ to fit into the GPU memory for larger models. For all training procedures, we select the last checkpoint for the comparison. When the pre-trained model was originally trained with extra data beyond the CIFAR-10 dataset, similar to Croce and Hein [2022], we use the extra 500k images introduced by Carmon et al. [2019] for fine-tuning, and each batch contains the same amount of standard and extra images. An epoch is completed when the whole standard training set has been used.

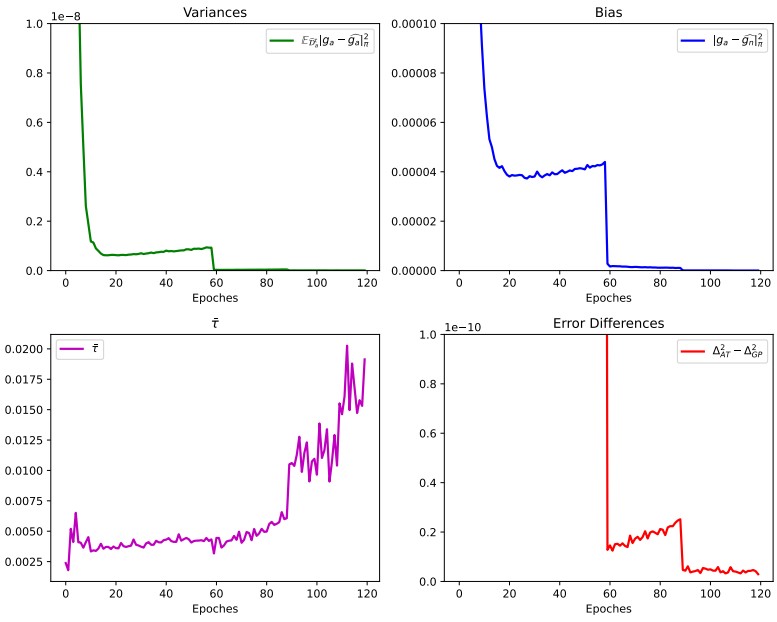

Figure 5: Plot of values of terms $E_{\widehat{D}_{a^t}} \|g_a - \widehat{g_a}\|_\pi^2$ (variance), $\|g_a - \widehat{g_n}\|_\pi^2$ (bias), $\bar{\tau}$, and $\Delta_{AT}^2 - \Delta_{GP}^2$ (error differences).

## B.2   Runtime Analysis of RAMP

We present runtime analysis results demonstrating the fact that RAMP is more expensive than E-AT and less expensive than MAX in Table 7. These results, recorded in seconds per epoch, were obtained using a single A100 40GB GPU. RAMP consistently supports that fact in all experiments.

Table 7: Analysis of time per epoch for RAMP and related baselines. RAMP is more expensive than E-AT and less expensive than MAX.

| Models \Methods | E-AT [Croce and Hein, 2022] | MAX | **RAMP** |
|---|---|---|---|
| CIFAR-10 RN-18 scratch | 78 | 219 | 157 |
| CIFAR-10 WRN-28-10 scratch | 334 | 1048 | 660 |
| CIFAR-10 RN-50 | 188 | 510 | 388 |
| CIFAR-10 WRN-34-20 | 1094 | 2986 | 2264 |
| CIFAR-10 WRN-28-10 carmon | 546 | 1420 | 1110 |
| CIFAR-10 WRN-28-10 gowal | 698 | 1895 | 1456 |
| CIFAR-10 WRN-70-16 | 3486 | 10330 | 7258 |
| ImageNet ResNet50 | 15656 | 41689 | 35038 |
| ImageNet Transformer | 38003 | 101646 | 81279 |

## B.3   Additional Results on RAMP Generalizing to Common Corruptions and Unseen Adversaries for Universal Robustness

In this section, we show **RAMP** can generalize better to other corruptions and unseen adversaries on union accuracy for stronger universal robustness.

**Implementations.** For the $l_0$ attack, we use Croce et al. [2022] with an epsilon of 9 pixels and $5k$ query points. For common corruptions, we directly use the implementation of RobustBench [Croce et al., 2020] for evaluation across 5 severity levels on all corruption types used in Hendrycks and Dietterich [2019]. For other unseen adversaries, we follow the implementation of Laidlaw et al. [2020], where we set $eps = 12$ for the fog attack, $eps = 0.5$ for the snow attack, $eps = 60$ for the gabor attack, $eps = 0.125$ for the elastic attack, and $eps = 0.125$ for the jpeglinf attack with 100 iterations. For ResNet-18 experiments, we do not compare with Winninghand [Diffenderfer et al.,

2021] since it uses a Wide-ResNet architecture. Also, we select the strongest baselines (E-AT and MAX) from the Wide-ResNet experiment results to compare for ResNet-18 experiments on universal robustness.

**Results.** For the ResNet-18 training from scratch experiment on CIFAR-10, in Table 8 and 9, we also show **RAMP** generally outperforms by $0.5\%$ on common corruptions and $7\%$ on union accuracy against unseen adversaries compared with E-AT.

Table 8: Accuracy against common corruptions using ResNet-18 on CIFAR-10 dataset.

| Models | common corruptions |
|--------|--------------------|
| E-AT | 73.8 |
| MAX | 75.1 |
| **RAMP** | 74.3 |

Table 9: Individual, average, and union accuracy against unseen adversaries using ResNet-18 on CIFAR-10 dataset.

| Models | $l_0$ | fog | snow | gabor | elastic | jpeglinf | Avg | Union |
|--------|-------|-----|------|-------|---------|----------|-----|-------|
| E-AT | 58.5 | 41.8 | 30.8 | 45.9 | 55.0 | 59.1 | 48.5 | 18.8 |
| MAX | 70.8 | 40.0 | 34.4 | 45.1 | 54.8 | 56.8 | 50.3 | 20.6 |
| **RAMP** | 56.8 | 40.5 | 40.5 | 50.0 | 59.2 | 56.2 | **50.5** | **25.9** |

## B.4 Additional Experiments with Different Epsilon Values

In this section, we provide additional results with different $\epsilon_1, \epsilon_2, \epsilon_\infty$ values. We select $\epsilon_\infty = [\frac{2}{255}, \frac{4}{255}, \frac{12}{255}, \frac{16}{255}]$, $\epsilon_1 = [6, 9, 12, 15]$, and $\epsilon_2 = [0.25, 0.75, 1.0, 1.5]$. We provide additional **RAMP** results compared with related baselines with training from scratch and performing robust fine-tuning in Section B.4.1 and Section B.4.2, respectively. We observe that **RAMP** can surpass E-AT with significant margins as well as a better accuracy-robustness tradeoff for both training from scratch and robust fine-tuning with $\lambda = 2.0$ for training from scratch and $\lambda = 0.5$ for robust fine-tuning in most cases.

### B.4.1 Additional Results with Training from Scratch

**Changing $l_\infty$ perturbations with $\epsilon_\infty = [\frac{2}{255}, \frac{4}{255}, \frac{12}{255}, \frac{16}{255}]$.** Table 10 and Table 11 show that **RAMP** consistently outperforms E-AT [Croce and Hein, 2022] on union accuracy when training from scratch.

Table 10: ($\epsilon_\infty = \frac{2}{255}, \epsilon_1 = 12, \epsilon_2 = 0.5$) and ($\epsilon_\infty = \frac{4}{255}, \epsilon_1 = 12, \epsilon_2 = 0.5$) with random initializations.

| | Clean | $l_\infty$ | $l_2$ | $l_1$ | Union | | Clean | $l_\infty$ | $l_2$ | $l_1$ | Union |
|--------|-------|------------|-------|-------|-------|------|-------|------------|-------|-------|-------|
| E-AT | 87.2 | 73.3 | 64.1 | 55.4 | 55.4 | E-AT | 86.8 | 58.9 | 66.4 | 54.6 | 53.7 |
| **RAMP** | 86.3 | 73.3 | 64.9 | 59.1 | **59.1** | **RAMP** | 86.1 | 60.0 | 67.4 | 58.5 | **57.4** |

Table 11: ($\epsilon_\infty = \frac{12}{255}, \epsilon_1 = 12, \epsilon_2 = 0.5$) and ($\epsilon_\infty = \frac{16}{255}, \epsilon_1 = 12, \epsilon_2 = 0.5$) with random initializations.

| | Clean | $l_\infty$ | $l_2$ | $l_1$ | Union | | Clean | $l_\infty$ | $l_2$ | $l_1$ | Union |
|--------|-------|------------|-------|-------|-------|------|-------|------------|-------|-------|-------|
| E-AT | 77.5 | 28.8 | 64.0 | 50.1 | 28.7 | E-AT | 69.4 | 18.8 | 58.7 | 47.7 | 18.7 |
| **RAMP** | 73.7 | 34.6 | 59.1 | 38.9 | **33.3** | **RAMP** | 65.0 | 25.7 | 49.8 | 32.6 | **25.0** |

**Changing $l_1$ perturbations with $\epsilon_1 = [6, 9, 12, 15]$.** Table 12 and Table 13 show that **RAMP** consistently outperforms E-AT [Croce and Hein, 2022] on union accuracy when training from scratch.

**Changing $l_2$ perturbations with $\epsilon_2 = [0.25, 0.75, 1.0, 1.5]$.** Table 14 and Table 15 show that **RAMP** consistently outperforms E-AT [Croce and Hein, 2022] on union accuracy when training from scratch.

Table 12: $(\epsilon_\infty = \frac{8}{255}, \epsilon_1 = 6, \epsilon_2 = 0.5)$ and $(\epsilon_\infty = \frac{8}{255}, \epsilon_1 = 9, \epsilon_2 = 0.5)$ with random initializations.

|  | Clean | $l_\infty$ | $l_2$ | $l_1$ | Union |  | Clean | $l_\infty$ | $l_2$ | $l_1$ | Union |
|---|---|---|---|---|---|---|---|---|---|---|---|
| E-AT | 85.5 | 43.1 | 67.9 | 63.9 | 42.8 | E-AT | 84.6 | 41.8 | 67.7 | 57.6 | 41.4 |
| **RAMP** | 83.8 | 48.1 | 63.0 | 51.2 | **46.0** | **RAMP** | 82.6 | 47.5 | 65.7 | 50.8 | **45.9** |

Table 13: $(\epsilon_\infty = \frac{8}{255}, \epsilon_1 = 15, \epsilon_2 = 0.5)$ and $(\epsilon_\infty = \frac{8}{255}, \epsilon_1 = 18, \epsilon_2 = 0.5)$ with random initializations.

|  | Clean | $l_\infty$ | $l_2$ | $l_1$ | Union |  | Clean | $l_\infty$ | $l_2$ | $l_1$ | Union |
|---|---|---|---|---|---|---|---|---|---|---|---|
| E-AT | 81.9 | 40.2 | 66.9 | 48.7 | 39.2 | E-AT | 81.0 | 39.8 | 65.8 | 44.3 | 38.0 |
| **RAMP** | 80.9 | 45.0 | 66.4 | 46.7 | **43.3** | **RAMP** | 79.9 | 43.5 | 65.7 | 45.0 | **41.9** |

Table 14: $(\epsilon_\infty = \frac{8}{255}, \epsilon_1 = 12, \epsilon_2 = 0.25)$ and $(\epsilon_\infty = \frac{8}{255}, \epsilon_1 = 12, \epsilon_2 = 0.75)$ with random initializations.

|  | Clean | $l_\infty$ | $l_2$ | $l_1$ | Union |  | Clean | $l_\infty$ | $l_2$ | $l_1$ | Union |
|---|---|---|---|---|---|---|---|---|---|---|---|
| E-AT | 82.8 | 41.3 | 75.6 | 52.9 | 40.5 | E-AT | 83.0 | 41.2 | 57.6 | 53.0 | 40.5 |
| **RAMP** | 81.8 | 46.0 | 74.7 | 48.8 | **44.5** | **RAMP** | 81.9 | 46.1 | 56.9 | 48.7 | **44.5** |

Table 15: $(\epsilon_\infty = \frac{8}{255}, \epsilon_1 = 12, \epsilon_2 = 1.0)$ and $(\epsilon_\infty = \frac{8}{255}, \epsilon_1 = 12, \epsilon_2 = 1.5)$ with random initializations.

|  | Clean | $l_\infty$ | $l_2$ | $l_1$ | Union |  | Clean | $l_\infty$ | $l_2$ | $l_1$ | Union |
|---|---|---|---|---|---|---|---|---|---|---|---|
| E-AT | 83.4 | 41.0 | 47.3 | 52.8 | 40.3 | E-AT | 83.5 | 41.0 | 25.5 | 52.9 | 25.5 |
| **RAMP** ($\lambda$=5) | 81.5 | 46.0 | 46.5 | 48.1 | **44.1** | **RAMP** | 74.4 | 43.4 | 37.2 | 51.1 | **37.1** |

## B.4.2 Additional Results with Robust Fine-tuning

**Changing $l_\infty$ perturbations with** $\epsilon_\infty = [\frac{2}{255}, \frac{4}{255}, \frac{12}{255}, \frac{16}{255}]$. Table 16 and Table 17 show that **RAMP** consistently outperforms E-AT [Croce and Hein, 2022] on union accuracy when performing robust fine-tuning.

Table 16: $(\epsilon_\infty = \frac{2}{255}, \epsilon_1 = 12, \epsilon_2 = 0.5)$ and $(\epsilon_\infty = \frac{4}{255}, \epsilon_1 = 12, \epsilon_2 = 0.5)$ with robust fine-tuning.

|  | Clean | $l_\infty$ | $l_2$ | $l_1$ | Union |  | Clean | $l_\infty$ | $l_2$ | $l_1$ | Union |
|---|---|---|---|---|---|---|---|---|---|---|---|
| E-AT | 86.5 | 74.8 | 66.7 | 57.9 | 57.9 | E-AT | 85.9 | 61.4 | 67.9 | 57.6 | 56.8 |
| **RAMP** | 85.8 | 74.0 | 66.2 | 60.1 | **60.1** | **RAMP** | 85.7 | 60.9 | 67.6 | 59.3 | **58.1** |

Table 17: $(\epsilon_\infty = \frac{12}{255}, \epsilon_1 = 12, \epsilon_2 = 0.5)$ and $(\epsilon_\infty = \frac{16}{255}, \epsilon_1 = 12, \epsilon_2 = 0.5)$ with robust fine-tuning.

|  | Clean | $l_\infty$ | $l_2$ | $l_1$ | Union |  | Clean | $l_\infty$ | $l_2$ | $l_1$ | Union |
|---|---|---|---|---|---|---|---|---|---|---|---|
| E-AT | 75.5 | 30.8 | 62.4 | 44.6 | 30.0 | E-AT | 68.7 | 20.7 | 56.1 | 42.1 | 20.5 |
| **RAMP** | 74.0 | 33.6 | 59.7 | 38.5 | **31.9** | **RAMP** | 65.6 | 25.0 | 51.5 | 31.2 | **23.8** |

**Changing $l_1$ perturbations with** $\epsilon_1 = [6, 9, 12, 15]$. Table 12 and Table 13 show that **RAMP** consistently outperforms E-AT [Croce and Hein, 2022] on union accuracy when performing robust fine-tuning.

**Changing $l_2$ perturbations with** $\epsilon_2 = [0.25, 0.75, 1.0, 1.5]$. Table 14 and Table 15 show that **RAMP** consistently outperforms E-AT [Croce and Hein, 2022] on union accuracy when performing robust fine-tuning.

Table 18: $(\epsilon_\infty = \frac{8}{255}, \epsilon_1 = \mathbf{6}, \epsilon_2 = 0.5)$ and $(\epsilon_\infty = \frac{8}{255}, \epsilon_1 = \mathbf{9}, \epsilon_2 = 0.5)$ with robust fine-tuning.

| | Clean | $l_\infty$ | $l_2$ | $l_1$ | Union | | Clean | $l_\infty$ | $l_2$ | $l_1$ | Union |
|---|---|---|---|---|---|---|---|---|---|---|---|
| E-AT | 84.2 | 45.8 | 66.8 | 59.0 | 45.0 | E-AT | 83.1 | 44.9 | 67.2 | 52.6 | 43.2 |
| **RAMP** | | | | | | **RAMP** | 82.5 | 47.1 | 66.0 | 49.9 | **44.8** |
| **($\lambda$=1.5)** | 83.0 | 48.7 | 63.5 | 51.7 | **46.4** | | | | | | |

Table 19: $(\epsilon_\infty = \frac{8}{255}, \epsilon_1 = \mathbf{15}, \epsilon_2 = 0.5)$ and $(\epsilon_\infty = \frac{8}{255}, \epsilon_1 = \mathbf{18}, \epsilon_2 = 0.5)$ with robust fine-tuning.

| | Clean | $l_\infty$ | $l_2$ | $l_1$ | Union | | Clean | $l_\infty$ | $l_2$ | $l_1$ | Union |
|---|---|---|---|---|---|---|---|---|---|---|---|
| E-AT | 81.3 | 43.5 | 66.6 | 42.8 | 39.0 | E-AT | 81.3 | 38.9 | 66.6 | 45.0 | 37.5 |
| **RAMP** | 80.4 | 44.2 | 66.1 | 44.4 | **41.2** | **RAMP** | 80.7 | 40.6 | 66.3 | 43.5 | **38.8** |

Table 20: $(\epsilon_\infty = \frac{8}{255}, \epsilon_1 = 12, \epsilon_2 = \mathbf{0.25})$ and $(\epsilon_\infty = \frac{8}{255}, \epsilon_1 = 12, \epsilon_2 = \mathbf{0.75})$ with robust fine-tuning.

| | Clean | $l_\infty$ | $l_2$ | $l_1$ | Union | | Clean | $l_\infty$ | $l_2$ | $l_1$ | Union |
|---|---|---|---|---|---|---|---|---|---|---|---|
| E-AT | 82.3 | 44.2 | 75.3 | 47.2 | 41.4 | E-AT | 83.0 | 43.5 | 58.1 | 46.5 | 40.4 |
| **RAMP** | 81.5 | 45.6 | 74.4 | 47.1 | **43.1** | **RAMP** | 81.4 | 45.6 | 57.4 | 47.2 | **42.9** |

Table 21: $(\epsilon_\infty = \frac{8}{255}, \epsilon_1 = 12, \epsilon_2 = \mathbf{1.0})$ and $(\epsilon_\infty = \frac{8}{255}, \epsilon_1 = 12, \epsilon_2 = \mathbf{1.5})$ with robust fine-tuning.

| | Clean | $l_\infty$ | $l_2$ | $l_1$ | Union | | Clean | $l_\infty$ | $l_2$ | $l_1$ | Union |
|---|---|---|---|---|---|---|---|---|---|---|---|
| E-AT | 82.3 | 41.0 | 49.0 | 51.6 | 40.2 | E-AT | 80.2 | 42.8 | 31.5 | 52.4 | 31.5 |
| **RAMP** | 81.4 | 45.6 | 47.8 | 47.1 | **42.9** | **RAMP** | 74.9 | 43.7 | 37.0 | 50.2 | **36.9** |

## B.5 Different Logit Pairing Methods

In this section, we test **RAMP** with robust fine-tuning using two more different logit pairing losses: (1) Mean Squared Error Loss ($\mathcal{L}_{mse}$) (Eq. 17), (2) Cosine-Similarity Loss ($\mathcal{L}_{cos}$) (Eq. 18). We replace the KL loss we used in the paper using the following losses. We use the same lambda value $\lambda = 1.5$ for both cases.

$$\mathcal{L}_{mse} = \frac{1}{n_c} \cdot \sum_{i=0}^{n_c} \frac{1}{2} \left(p_q[\gamma[i]] - p_r[\gamma[i]]\right)^2 \tag{17}$$

$$\mathcal{L}_{cos} = \frac{1}{n_c} \cdot \sum_{i=0}^{n_c} \left(1 - \cos(p_q[\gamma[i]], p_r[\gamma[i]])\right) \tag{18}$$

Table 22 displays **RAMP** robust fine-tuning results of different logit pairing losses using PreAct-ResNet-18 on CIFAR-10 with $\lambda = 1.5$. We see those losses generally improve union accuracy compared with baselines in Table 24. $\mathcal{L}_{cos}$ has a better clean accuracy yet slightly worsened union accuracy. $\mathcal{L}_{mse}$ has the best union accuracy and the worst clean accuracy. $\mathcal{L}_{KL}$ is in the middle of the two others. However, we acknowledge the possibility that each logit pairing loss may have its own best-tuned $\lambda$ value.

Table 22: **RAMP** fine-tuning results of different logit pairing losses using PreAct-ResNet-18 on CIFAR-10.

| Losses | Clean | $l_\infty$ | $l_2$ | $l_1$ | Union |
|---|---|---|---|---|---|
| KL | 80.9 | **45.5** | 66.2 | 47.3 | 43.1 |
| MSE | 80.4 | **45.6** | 65.8 | **47.6** | **43.5** |
| Cosine | **81.6** | 45.4 | **66.7** | 47.0 | 42.9 |

## B.6 AT from Scratch Using WideResNet-28-10

**Implementations.** We use a cyclic learning rate with a maximum rate of $0.1$ for 30 epochs and adopt the outer minimization trades loss from Zhang et al. [2019] with the default hyperparameters, same as Croce and Hein [2022]; also, we set $\lambda = 2.0$ and $\beta = 0.5$ for training **RAMP**. Additionally, we use the WideResNet-28-10 architecture same as Zagoruyko and Komodakis [2016] for our reimplementations on CIFAR-10.

**Results.** Since the implementation of experiments on WideResNet-28-10 in Croce and Hein [2022] paper is not public at present, we report our implementation results on E-AT, where our results show that **RAMP** outperforms E-AT in union accuracy with a significant margin, as shown in Table 23. Also, we experiment with using the trade loss (**RAMP w trades**) for the outer minimization, we observe that **RAMP w trades** achieves a better union accuracy at the loss of some clean accuracy.

Table 23: **WideResNet-28-10 trained from random initialization** on CIFAR-10. **RAMP** outperforms E-AT on union accuracy with our implementation.

| Methods | Clean | $l_\infty$ | $l_2$ | $l_1$ | Union |
|---|---|---|---|---|---|
| E-AT w trades (reported in Croce and Hein [2022]) | 79.9 | 46.6 | 66.2 | 56.0 | 46.4 |
| E-AT w trades (ours) | 79.2 | 44.2 | 64.9 | 54.9 | 44.0 |
| **RAMP w/o trades** (ours) | 81.1 | 46.6 | 65.9 | 48.1 | 44.6 |
| **RAMP w trades** (ours) | 79.9 | 47.1 | 65.1 | 49.0 | **45.8** |

## B.7 Robust Fine-tuning Using PreAct-ResNet-18

**Implementations.** For robust fine-tuning with ResNet-18, we perform 3 epochs on CIFAR-10. We set the learning rate as $0.05$ for PreAct-ResNet-18 and $0.01$ for other models. We set $\lambda = 0.5$ in this case. Also, we reduce the learning rate by a factor of 10 after completing each epoch.

**Result.** Table 24 shows the robust fine-tuning results using PreAct ResNet-18 model on the CIFAR-10 dataset with different methods. The results for all baselines are directly from the E-AT paper [Croce and Hein, 2022] where the authors reimplemented other baselines (e.g., MSD, MAX) to achieve better union accuracy than presented in the original works. **RAMP** surpasses all other methods on union accuracy.

Table 24: **RN-18 $l_\infty$-AT model fine-tuned** for 3 epochs (repeated for 5 seeds). **RAMP** has the highest union accuracy. Baseline results are from Croce and Hein [2022].

| Methods | Clean | $l_\infty$ | $l_2$ | $l_1$ | Union |
|---|---|---|---|---|---|
| RN-18- $l_\infty$-AT | 83.7 | 48.1 | 59.8 | 7.7 | 38.5 |
| + SAT | $83.5 \pm 0.2$ | $43.5 \pm 0.2$ | $68.0 \pm 0.4$ | $47.4 \pm 0.5$ | $41.0 \pm 0.3$ |
| + AVG | $84.2 \pm 0.4$ | $43.3 \pm 0.4$ | $68.4 \pm 0.6$ | $46.9 \pm 0.6$ | $40.6 \pm 0.4$ |
| + MAX | $82.2 \pm 0.3$ | $45.2 \pm 0.4$ | $67.0 \pm 0.7$ | $46.1 \pm 0.4$ | $42.2 \pm 0.6$ |
| + MSD | $82.2 \pm 0.4$ | $44.9 \pm 0.3$ | $67.1 \pm 0.6$ | $47.2 \pm 0.6$ | $42.6 \pm 0.2$ |
| + E-AT | $82.7 \pm 0.4$ | $44.3 \pm 0.6$ | $68.1 \pm 0.5$ | $48.7 \pm 0.5$ | $42.2 \pm 0.8$ |
| + **RAMP** ($\lambda$=1.5) | $81.1 \pm 0.2$ | $45.4 \pm 0.3$ | $66.1 \pm 0.2$ | $47.2 \pm 0.1$ | **$43.1 \pm 0.2$** |
| + **RAMP** ($\lambda = 0.5$) | $81.5 \pm 0.1$ | $45.5 \pm 0.2$ | $66.4 \pm 0.2$ | $47.0 \pm 0.1$ | **$42.9 \pm 0.2$** |

## B.8 Robust Fine-tuning with More Epochs

In Table 25, we apply robust fine-tuning on the PreAct ResNet-18 model for the CIFAR-10 dataset with $5, 7, 10, 15$ epochs, and compare it with E-AT. **RAMP** consistently outperforms the baseline on union accuracy, with a larger improvement when we increase the number of epochs.

# C  Additional Visualization Results

In this section, we provide additional t-SNE visualizations of the multiple-norm tradeoff and robust fine-tuning procedures using different methods.

Table 25: **Fine-tuning with more epochs**: **RAMP** consistently outperforms E-AT on union accuracy. E-AT results are from Croce and Hein [2022].

| | 5 epochs | | 7 epochs | | 10 epochs | | 15 epochs | |
| --- | --- | --- | --- | --- | --- | --- | --- | --- |
| | Clean | Union | Clean | Union | Clean | Union | Clean | Union |
| E-AT | 83.0 | 43.1 | 83.1 | 42.6 | 84.0 | 42.8 | 84.6 | 43.2 |
| **RAMP** | 81.7 | **43.6** | 82.1 | **43.8** | 82.5 | **44.6** | 83.0 | **44.9** |

## C.1 Pre-trained $l_1, l_2, l_\infty$ AT models

Figure 6 shows the robust accuracy of $l_1, l_2, l_\infty$ AT models against their respect $l_1, l_2, l_\infty$ perturbations, on CIFAR-10 using PreAct-ResNet-18 architecture. Similar to Figure **??**, $l_\infty$-AT model has a low $l_1$ robustness and vice versa. In this common choice of epsilons, we further confirm that $l_\infty - l_1$ is the key trade-off pair.

## C.2 Robust Fine-tuning for all Epochs

We provide the complete visualizations of robust fine-tuning for 3 epochs on CIFAR-10 using $l_1$ examples, E-AT, and **RAMP**. Rows in $l_1$ fine-tuning (Figure 7), E-AT fine-tuning (Figure 8), and **RAMP** fine-tuning (Figure 9) show the robust accuracy against $l_\infty, l_1, l_2$ attacks individually, of epoch $0, 1, 2, 3$, respectively. We observe that throughout the procedure, **RAMP** manages to maintain more $l_\infty$ robustness during the fine-tuning with more points colored in cyan, in comparison with two other methods. This visualization confirms that after we identify a $l_p - l_r (p \neq r)$ key tradeoff pair, **RAMP** successfully preserves more $l_p$ robustness when training with some $l_r$ examples via enforcing union predictions with the logit pairing loss.

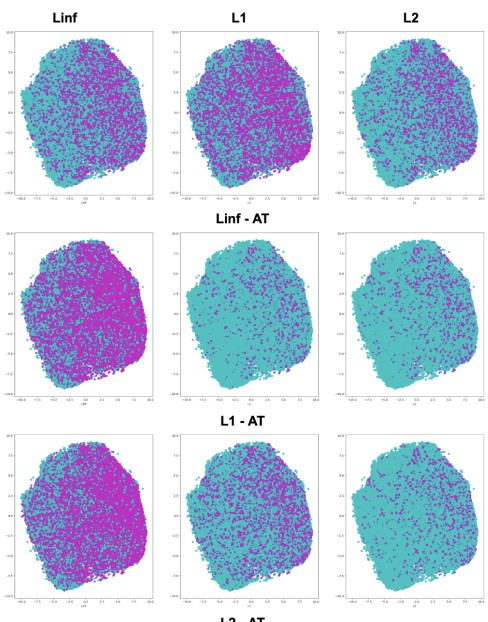

Figure 6: $l_1, l_2, l_\infty$ pre-trained RN18 $l_\infty$-AT models with correct/incorrect predictions against $l_1, l_2, l_\infty$ attacks. Correct predictions are colored with cyan and incorrect with magenta. Each row represents $l_\infty, l_1, l_2$ AT models, respectively. Each column shows the accuracy concerning a certain $l_p$ attack.

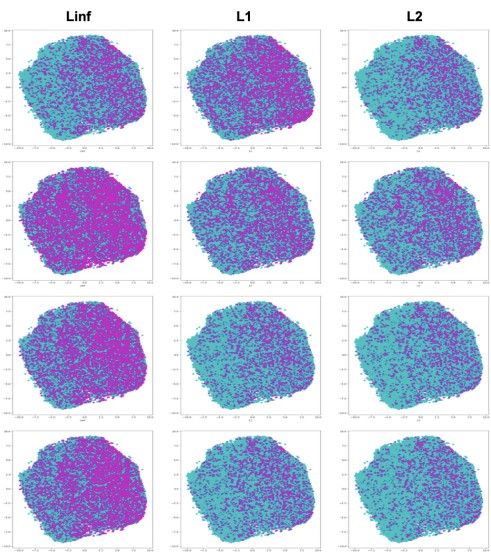

Figure 7: **Finetune RN18 $l_\infty$-AT model on $l_1$ examples for 3 epochs**. Each row represents the prediction results of epoch $0, 1, 2, 3$ respectively.

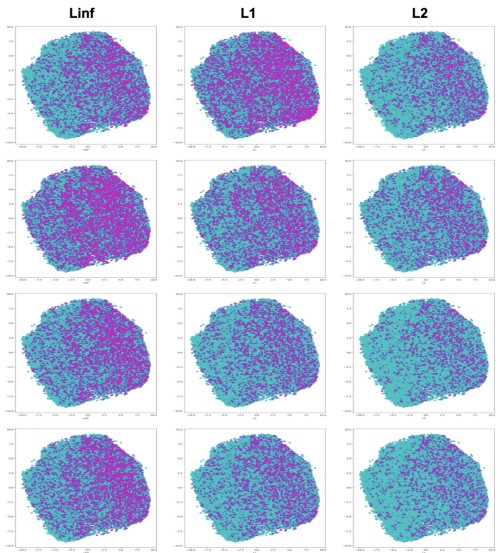

Figure 8: **Finetune RN18 $l_\infty$-AT model with E-AT for 3 epochs**. Each row represents the prediction results of epoch $0, 1, 2, 3$ respectively.

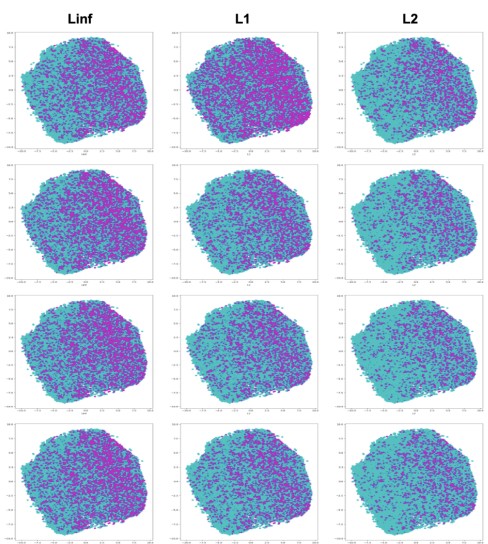

Figure 9: **Finetune RN18 $l_\infty$-AT model with RAMP for 3 epochs**. Each row represents the prediction results of epoch $0, 1, 2, 3$ respectively.

