# OpenReview forum: "RAMP: Boosting Adversarial Robustness Against Multiple $l_p$ Perturbations for Universal Robustness"
_NeurIPS.cc/2024/Conference — NeurIPS 2024 poster_

### Official Review · Reviewer_Qhq5 · 2024-07-01

**Soundness:** 3
**Presentation:** 2
**Contribution:** 2
**Rating:** 6
**Confidence:** 3

**Summary:**

This paper addresses the challenge of training Machine Learning (ML) models that show good robustness against multiple l_p attacks and accuracy. This problem has already been studied in previous work [33,6], but the papers discuss the problem from the lens of distribution shift. This point of view represents the starting point for proposing a new adversarial training (AT) algorithm called RAMP based on two building blocks: (i) a new logit pairing loss that allows a model robust against a l_p norm to exhibit high robustness w.r.t. another l_q norm (with q \neq p); (ii) a new model parameter update based on the combinations of model updates from natural training and adversarial training to train a model that observes a better accuracy/union robustness tradeoff that SOTA AT models. A deep theoretical analysis shows that, for large models and a reasonable choice of parameter values, the novel parameter update allows the adversarial training algorithm to converge faster, training a model with higher robustness and accuracy. The extensive experimental evaluation highlights that the trained large models on ImageNet and Cifar-10 exhibit a better accuracy-union robustness tradeoff than SOTA AT proposals [6].

**Strengths:**

**Original AT algorithm to address the accuracy-union robustness trade-off**: the novel AT training addresses both the problem of improving the robustness on multiple l_p perturbations and improving the accuracy-robustness trade-off at the same time by exploiting a novel connection with natural training. Addressing the two problems at the same time constitutes the novelty of the work.

**Theoretical analysis of the impact of gradient projection on adversarial robustness**: the deep theoretical analysis shows that integrating model parameter updates from natural training in the adversarial training updates allows the training to converge better for large models.

**Comprehensive experimental settings considered**: the experimental evaluation is performed on full training and fine-tuning large models used by relevant competitors in the literature [6], on significant datasets (CIFAR-10 and ImageNet) and considering several combinations of perturbation values for l_inf, l_1, and l_2.

**Weaknesses:**

**Analysis of time per epoch is not opportunely deepened (addressed by the authors)**: the analysis regarding the computational time per epoch is limited since the results are shown for only one model (ResNet-18) on only one dataset (CIFAR-10), but seven other models are employed in the experimental evaluation. Since a lot of additional results have been included in the Appendix, I would have expected to see also more results supporting the statement ``` RAMP has around two times the runtime cost of E-AT and 2/3 cost of MAX``` (Section 5.2). For example, the authors may want to report the time per epoch on AT the ResNet-18 and the WRN-28-10 from scratch on CIFAR-10 (as reported in [6]) and on training another model on ImageNet.

**Trade-off between robustness and accuracy seems to not be properly addressed**: Even though RAMP allows the model to obtain better union accuracy, the clean accuracy is in any case lower than the clean accuracy obtained by using the baseline E-AT [6]. As an example, the RAMP fine-tuned RN-50-$\ell_\infty$ shows a clean accuracy that is 3.8 points lower than the one obtained by the E-AT fine-tuned model and a gain in union accuracy that is one point (Table 1). Moreover, more than half of the results shown in the Appendix and some results in the main body show that the gain in the union accuracy is obtained at the expense of a comparable loss in the clean accuracy. If the trade-off between robustness and accuracy had been properly addressed, I would have expected to see a comparable clean accuracy and better union accuracy than the one obtained by E-AT. These results limit the novelty of RAMP since it seems to be slower than E-AT and only allows the model to obtain better union accuracy without properly addressing the accuracy/robustness trade-off. I suggest that the authors explain why these cases happen and how to improve also the accuracy of the obtained model.

**Questions:**

- Can the authors show more results supporting the fact that the time per epoch required by RAMP is partially longer than the time per epoch required by E-AT? (See the Weaknesses section for more details)

- Is it possible to use RAMP to obtain an accuracy close to the one obtained by using E-AT and a better union accuracy? (See the Weaknesses section for more details)

**Updates after authors' response**: The authors provided additional results to address the first question, effectively addressing the first weakness. After discussion, the answer to the second question is more convincing. RAMP seems to address the accuracy-robustness trade-off better than E-AT even by choosing small values of $\lambda$, as the authors wrote. So, choosing a small value of $\lambda$ helps preserve the accuracy while still outperforming the models trained with E-AT regarding union accuracy. This choice could be right in general.

Reasons for Not Giving a Higher Score: RAMP does not allow the user to obtain an impressive gain in union accuracy as E-AT did when the corresponding paper was published. However, it allows the model to obtain a few points more of union accuracy while preserving an accuracy similar to that of models trained with E-AT. I can say that this paper will have a moderate to high impact, but I am not convinced that it will have a high impact for sure.

Reasons for Not Giving a Lower Score: In my opinion, the work is solid from a theoretical and methodological point of view. The experimental results are more convincing after the discussion with the authors.

**Limitations:**

The authors have explained the limitations in Section 5.2 in brief. However, the limitations should be addressed better:
- The analysis of the computational cost should be supported with more experimental results.
- The authors should better substantiate the sentence ``` We notice occasional drops in clean accuracy during fine-tuning with RAMP```. Indeed, the drop in accuracy is exhibited also when the model is trained from scratch (see Section B.2.1 in the Appendix). Moreover, the gain in the union accuracy is often comparable to the loss in clean accuracy. Thus, the accuracy/robustness trade-off seems not to be properly addressed.

This work has no negative societal and ethical impacts since it contributes to improving the robustness of AI systems in practice. The experimental and implementation details have been documented.

---

> ### Author Rebuttal · Authors · 2024-08-06
>
> - **Analysis of time per epoch**: We present additional results demonstrating the fact that RAMP is more expensive than E-AT and less expensive than MAX. These results, recorded in seconds per epoch, were obtained using a single A100 GPU. RAMP consistently supports that fact in all experiments.
>
> | Models \ Methods         | E-AT  | MAX    | RAMP  |
> |--------------------------|-------|--------|-------|
> | CIFAR RN-18 scratch      | 78    | 219    | 157   |
> | CIFAR WRN-28-10 scratch  | 334   | 1048   | 660   |
> | CIFAR RN-50              | 188   | 510    | 388   |
> | CIFAR WRN-34-20          | 1094  | 2986   | 2264  |
> | CIFAR WRN-28-10 carmon   | 546   | 1420   | 1110  |
> | CIFAR WRN-28-10 gowal    | 698   | 1895   | 1456  |
> | CIFAR WRN-70-16          | 3486  | 10330  | 7258  |
> | ImageNet ResNet50        | 15656 | 41689  | 35038 |
> | ImageNet Transformer     | 38003 | 101646 | 81279 |
>
> - **The trade-off between robustness and accuracy**:
>
> **It is possible to use RAMP to obtain an accuracy close to the one obtained by E-AT and a better union accuracy by setting a small $\lambda$ value**. The clean accuracy drops because our main goal is to improve union accuracy. To improve the clean accuracy of the obtained model, we can set a smaller $\lambda$ value (as shown in Figure 4(a) (b)). For instance, in Figure 4(b), when $\lambda=1$, RAMP has a clean accuracy of 81.9% and union accuracy of **44.4%**, compared with E-AT's 82.2% and 42.4%. Thus, one can usually achieve an accuracy close to E-AT and a better robustness by setting a relatively small $\lambda$ value.
>
> Further, we show as follows that **RAMP obtains a better robustness and accuracy tradeoff by generalizing better to common corruptions and other unseen adversaries, and the overall gain in robustness is more than the drop in accuracy**. Compared with SOTA common corruption model [2] and other pretrained $l_p$ models on wideresnet-28-10 experiment training from scratch on CIFAR-10, RAMP shows the best tradeoff between robustness and accuracy against various kinds of perturbations and corruptions.
>
> For common corruptions, RAMP achieves a 4% improvement compared to E-AT and MAX, averaged across five severity levels and all common corruptions.
>
> | model           | common corruptions |
> |-----------------|--------------------|
> | l1-AT           |               78.2 |
> | l2-AT           |               77.2 |
> | linf-AT         |               73.4 |
> | winninghand [2]  |           **91.1** |
> | wideresnet E-AT  |               71.5 |
> | wideresenet MAX |                 71 |
> | wideresnet RAMP |               75.5 |
>
>
> For other attack types, such as $l_0$ and other unseen adversaries, RAMP shows the best average and union accuracy, with a 4% improvement over E-AT and a 2% improvement over MAX.
>
> | model           | l0 (eps=9,query=5k) | fog (eps=128) | snow (eps=0.5) | gabor (eps=60) |  elastic (eps=0.125) | jpeg(eps=0.125) | avg      | union    |
> |-----------------|---------------------|---------------|----------------|----------------|----------------------|-----------------|----------|----------|
> | l1-AT           |                  79 |          41.4 |           22.9 |           40.5 |                 48.9 |            48.4 |     46.9 |     12.8 |
> | l2-AT           |                67.5 |          48.7 |           26.1 |           44.1 |                 53.2 |            45.4 |     47.5 |     16.2 |
> | linf-AT         |                55.5 |          44.7 |           32.9 |           53.8 |                 56.6 |            33.4 |     46.2 |     19.1 |
> | winninghand [2]  |                74.1 |          74.5 |           18.3 |           76.5 |                 12.6 |               0 |     42.7 |        0 |
> | wideresnet E-AT  |                58.5 |          35.9 |           35.3 |           50.7 |                 55.7 |            60.3 |     49.4 |     21.9 |
> | wideresenet MAX |                56.2 |          42.9 |           35.4 |           49.8 |                 57.8 |            55.7 |     49.6 |     24.4 |
> | wideresnet RAMP |                55.5 |          40.5 |           40.2 |           52.9 |                 60.3 |            56.1 | **50.9** | **26.1** |
>
> For Resnet-18 training from scratch experiment on CIFAR-10, we also show RAMP outperforms by 0.5% on common corruptions and 7% on union accuracy against unseen adversaries compared with E-AT.
> | model | common corruptions |
> |-------|--------------------|
> | E-AT   |               73.8 |
> | RAMP  |           **74.3** |
>
> | model | l0 (eps=9,query=5k) | fog (eps=128) | snow (eps=0.5) | gabor (eps=60) |  elastic (eps=0.125) | jpeg(eps=0.125) | avg      | union    |
> |-------|---------------------|---------------|----------------|----------------|----------------------|-----------------|----------|----------|
> | E-AT   |                58.5 |          41.8 |           30.8 |           45.9 |                   55 |            59.1 |     48.5 |     18.8 |
> | RAMP  |                56.8 |          40.5 |           40.5 |             50 |                 59.2 |            56.2 | **50.5** | **25.9** |
>
> These results highlight that RAMP effectively enhances model robustness across diverse adversarial scenarios and attains a better robustness/accuracy tradeoff evaluated on various kinds of perturbations and corruptions.
>
>
> [1] Adversarial Robustness against Multiple and Single lp-threat Models via Quick Fine-Tuning of Robust Classifiers.
>
> [2] A Winning Hand: Compressing Deep Networks Can Improve Out-Of-Distribution Robustness.

---

> > ### Author Response · Authors · 2024-08-09
> > **follow up**
> >
> > Dear reviewer,
> >
> > We want to thank you again for your efforts in reviewing our submission.
> >
> > We have tried our best to respond to your questions. If our responses address your concerns, we sincerely hope you can reconsider the scores. We will also be very happy to answer any follow-up questions. Look forward to your updates.
> >
> > Thanks,
> > Authors

---

> ### Comment · Reviewer_Qhq5 · 2024-08-09
> **Thanks for your response, I have other concerns**
>
> I thank the authors for their response.
>
> The answer to the first question is clear, thanks to the additional results provided. I strongly encourage you to include these results in the next version of the paper.
>
> The response to the second question is interesting. Noting that 'RAMP obtains a better robustness and accuracy tradeoff by generalizing better to common corruptions and other unseen adversaries' is certainly another valuable contribution. Additionally, I appreciate your emphasis on the importance of the $\lambda$ value in achieving the desired accuracy-robustness trade-off. However, I am somewhat concerned that $\lambda$ may need to be fine-tuned to reach the desired trade-off, as we lack clear evidence that the union accuracy will decrease as minimally as shown in Figure 4 when moving to lower $\lambda$ values. Indeed, that figure refers to only one model, and we have no data on the impact of $\lambda$ when training other models. Fine-tuning can require a lot of computational time, since you shown that training with RAMP may require more than twice the time required by E-AT. Moreover, why did you not consistently choose a smaller $\lambda$ value, even when training from scratch (you set $\lambda=5$ in the paper), to obtain accuracy comparable to E-AT while consistently achieving higher robustness?
>
> To sum up, your analysis has improved my opinion of your work, but there is insufficient evidence regarding the influence of the $\lambda$ value on clean and union accuracy across multiple models to say that the accuracy-robustness trade-off is properly addressed.

---

> ### Author Response · Authors · 2024-08-12
> **Thank you and response to choose lambda values**
>
> We thank the reviewer for their detailed reply. We will include the time cost analysis in the next version of the paper. Also, here we provide more results on $\lambda$ value experiments. RAMP can achieve a better robustness-accuracy tradeoff compared with E-AT by choosing consistent $\lambda$ values for multiple models. Here we set $\lambda=0.5$ for robust fine-tuning. Also, we are running from scratch experiments and will report as soon as we have the results.
>
> |        Models        | Methods | Clean | linf |  l2  |  l1  |   union  |
> |:--------------------:|:-------:|:-----:|:----:|:----:|:----:|:--------:|
> |         RN-50        |   E-AT  |  85.3 | 46.5 | 68.3 | 45.3 |   41.6   |
> |                      |   RAMP  |  84.3 | 47.0 | 67.7 | 46.5 | **43.3** |
> |       WRN-34-20      |   E-AT  |  87.8 | 49.0 | 71.6 | 49.8 |   45.1   |
> |                      |   RAMP  |  87.1 | 49.7 | 70.8 | 50.4 | **46.9** |
> | WRN-28-10 (*) carmon |   E-AT  |  89.3 | 51.8 | 74.6 | 53.3 |   47.9   |
> |                      |   RAMP  |  89.2 | 55.9 | 74.7 | 55.7 | **52.7** |
> |  WRN-28-10 (*) gowal |   E-AT  |  89.8 | 54.4 | 76.1 | 56.0 |   50.5   |
> |                      |   RAMP  |  89.4 | 55.9 | 74.7 | 56.0 | **52.9** |
> |     WRN-70-16 (*)    |   E-AT  |  89.6 | 54.4 | 76.7 | 58.0 |   51.6   |
> |                      |   RAMP  |  90.6 | 54.7 | 74.6 | 57.9 | **53.3** |

---

> > ### Comment · Reviewer_Qhq5 · 2024-08-12
> > **Thanks for the additional results you are providing**
> >
> > I greatly appreciate your dedication in showing that RAMP can consistently outperform E-AT when training from scratch, even with a small $\lambda$, to be coherent with what you wrote.
> >
> > I have only one question about the data. I suppose you are producing the data in Table 1 again. Why are the results obtained with E-AT different from the results you show in Table 1 in the paper? Are you training the models again using E-AT to ensure that they are all trained in the same setting?

---

> ### Author Response · Authors · 2024-08-12
>
> Thanks for the quick response. The numbers are different because we regenerated both E-AT (we trained them again) and RAMP results testing on all testing data points instead of the 1000 points in the paper, as suggested by Reviewer jXrr (see our response to Reviewer jXrr).

---

> > ### Comment · Reviewer_Qhq5 · 2024-08-12
> > **Official Comment by Reviewer Qhq5**
> >
> > Thanks also to you for the quick response. Since you have provided more evidence that the small values of $\lambda$ allow RAMP to outperform E-AT, as you previously wrote, I am more confident now that RAMP can address better than E-AT the accuracy-robustness trade-off. Thus, I will increase my score. I invite you to update the results in the next version of your paper.

---

> > > ### Author Response · Authors · 2024-08-13
> > > **Thank you and we provide additional results**
> > >
> > > Thank you very much for considering our work and for increasing the score. We will incorporate the updated results into the next version of our paper. In the meantime, we would like to share additional results that demonstrate improved tradeoffs with smaller $\lambda$ values compared to E-AT (training from scratch, $\lambda$ = 2).
> > >
> > > |      Models      | Methods | Clean | linf |  l2  |  l1  | union |
> > > |:----------------:|:-------:|:-----:|:----:|:----:|:----:|:-----:|
> > > |     resnet-18    |   E-AT  |  82.2 | 42.7 | 67.5 | 53.6 |  42.4 |
> > > |                  |   RAMP  |  81.9 | 46.1 | 66.6 | 48.9 |  44.4 |
> > > | wideresnet-28-10 |   E-AT  |  79.2 | 44.2 | 64.9 | 54.9 |  44.0 |
> > > |                  |   RAMP  |  84.5 | 45.5 | 67.3 | 47.8 |  44.0 |

---

### Official Review · Reviewer_3Hj9 · 2024-07-05

**Soundness:** 2
**Presentation:** 3
**Contribution:** 2
**Rating:** 3
**Confidence:** 4

**Summary:**

The paper proposes a framework called RAMP to boost the adversarial robustness against multiple $l_p$ perturbations. Compared with existing methods, RAMP can better retain the robustness against the original type of perturbations when fine-tuning the model. The authors provide theoretical and empirical justifications to demonstrate the effectiveness of RAMP.

**Strengths:**

++ The targeted problem is well-motivated and formulated.

++ The experimental results are comprehensive with reliable evaluation metrics.

++ Theoretical justifications can better explain the effectiveness of the proposed methods.

**Weaknesses:**

Despite the strength in the last section, I have the following concerns:

1. Novelty in the algorithm: logit pairing is not a new idea. Actually, in Equation (1), if we replace $p_r$ with the probability of the clean input, the algorithm will become TRADES. Some findings, such as trade-offs between accuracy and robustness, are not new.

2. Generality of the algorithm: the manuscript only studies the joint robustness among $l_1$, $l_2$ and $l_\infty$ perturbations, especially the $l_1$ and $l_\infty$ ones. This is too limited. The authors should consider more generic cases, such as $l_0$ bounded perturbations and common corruptions (Gaussian noise, color re-mapping e.t.c.). I believe this will make the algorithm more generally applicable.

3. Novelty and precision of the theorem: the theoretical tools employed to analyze convergence in Theorem 4.5 are standard and not unique. In Theorem 4.6, if $\beta = 0$, $\widehat{g}\_a$ should be similar to $g_{GP}$, so $\Delta_{AT}$ should be very close to $\Delta_{GP}$. Furthermore, the left hand side of the approximation should be close to $0$, can the author explain how the right hand side approximation reflects this? In addition, using $\simeq$ is not rigorous, the authors should at least point out the order of the difference between the left hand side and the right hand side.

**Questions:**

The weakness part points out the major questions I would ask. Below are additional relatively minor questions:

1. Why don't use AutoAttack as the evaluation metric in Figure 2?

2. In Figure 4, what is the difference between MAX and RAMP with $\lambda = 0$?

My major concern of this paper is about the novelty, because both the motivation (logit pairing) and many parts of the analysis are standard. In addition, the algorithm is not generic, as it is only evaluated under $l_1$, $l_2$ and $l_\infty$ bounded perturbations. Therefore, I cannot recommend acceptance at this point. I welcome the authors to address my concerns in the rebuttal.

**Limitations:**

The limitations and the societal impact are briefly discussed in the end of the paper.

---

> ### Author Rebuttal · Authors · 2024-08-06
>
> - **Novelty in the algorithm**: Firstly, $p_r$ represents the predictions of $l_r$ adversarial examples, not the probability of the clean input. Our logit pairing method differs from TRADES as we only regularize the logits on the correctly predicted $l_r$ adversarial example subset, which TRADES does not consider. This approach enhances union accuracy by aligning the logit distributions of $l_r$ and $l_q$ attacks on the correctly predicted $l_r$ subset. Additionally, in Appendix Table 18, we demonstrate that combining RAMP with TRADES loss improves union accuracy to 45.2% (up from 44.6%) in the WideResNet-28-10 experiment training from scratch. Secondly, Our gradient projection method is more generalizable than traditional methods mitigating the trade-off between accuracy and robustness, such as TRADES [1], MART [2], and HAT [3]. Previous works limit to a certain type of perturbation for better robustness-accuracy tradeoff. However, our gradient projection method is effective for both single (Figures 2 and 3) and multiple perturbations/corruptions (Table 2 and 3, also see in our rebuttal to Reviewer Qhq5).
>
> - **Generality of the algorithm**:
> Training for multiple norm robustness allows generalization beyond the threat model used during training [1]. Our method achieves superior average and union accuracy compared to the SOTA common corruption model [5], $l_p$ pretrained models, E-AT, MAX across other perturbations beyond multiple-norm perturbations. Specifically, it obtains good accuracy against common corruptions with five severity levels and shows the best average and union accuracy performance against multiple unseen adversaries. Using WideResNet 28-10 models, RAMP achieves a 4% improvement over E-AT and MAX on common corruptions, averaged across five severity levels. For other attack types, such as $l_0$ and unseen adversaries, RAMP shows the best average and union accuracy, with a 4% improvement over E-AT and 2% over MAX. These results highlight that RAMP effectively enhances model robustness across diverse adversarial scenarios.
>
> | model           | common corr. | l0 (eps=9,query=5k) | fog (eps=128) | snow (eps=0.5) | gabor (eps=60) |  elastic (eps=0.125) | jpeg(eps=0.125) | avg      | union    |
> |-----------------|--------------------|---------------------|---------------|----------------|----------------|----------------------|-----------------|----------|----------|
> | l1-AT           |               78.2 |                  79 |          41.4 |           22.9 |           40.5 |                 48.9 |            48.4 |     46.9 |     12.8 |
> | l2-AT           |               77.2 |                67.5 |          48.7 |           26.1 |           44.1 |                 53.2 |            45.4 |     47.5 |     16.2 |
> | linf-AT         |               73.4 |                55.5 |          44.7 |           32.9 |           53.8 |                 56.6 |            33.4 |     46.2 |     19.1 |
> | winninghand [5]  |           **91.1** |                74.1 |          74.5 |           18.3 |           76.5 |                 12.6 |               0 |     42.7 |        0 |
> | E-AT  |               71.5 |                58.5 |          35.9 |           35.3 |           50.7 |                 55.7 |            60.3 |     49.4 |     21.9 |
> | MAX |                 71 |                56.2 |          42.9 |           35.4 |           49.8 |                 57.8 |            55.7 |     49.6 |     24.4 |
> | RAMP |               75.5 |                55.5 |          40.5 |           40.2 |           52.9 |                 60.3 |            56.1 | **50.9** | **26.1** |
>
> - **Novelty and precision of the theorem**:
>
> The novelty of Theorem 4.5 lies in its consideration of the changing adversarial example distribution $D_{a^t}$ at each time step $t$ to get the convergence proof. To sum over $t=0,\dots,T-1$, we perform some relaxations to obtain the summation for changing distrbutions. Also, for Equation 14, we bound the left-hand side with the maximum $\|\widehat{g_{\texttt{Aggr}}}(\widehat{\theta_{\texttt{Aggr}}})-\nabla L_{D_{a^t}}(\widehat{\theta_{\texttt{Aggr}}})\|^2$ one can get during the optimization steps. Theorem 4.5 links the convergence of the aggregation rule with the delta error analysis in Theorem 4.6. Specifically, Theorem 4.6 demonstrates that a smaller delta error results in better convergence with a tighter bound.
>
> Further, we have carefully checked the theorem and Theorem 4.6 will be updated as follows:
>
> $\Delta^2_{AT} -\Delta^2_{GP} \approx \beta (2 - \beta) E_{\widehat{D_{a^t}}} \|g_{a} -\widehat{g_{a}} \|^2_\pi - \beta^2 \bar\tau^2 \|g_{a} - \widehat{g_{n}}\|^2_\pi$
>
> When $\beta = 0$, the difference between two aggregation errors becomes 0. Further, we estimate the actual values of terms $E_{\widehat{D_{a^t}}} \|g_{a} - \widehat{g_{a}} \|^2_\pi$ (variance), $\|g_{a} -\widehat{g_{n}}\|^2_\pi$ (bias), and $\bar\tau$ using the estimation methods in [4]. We show across different epochs, the delta error difference **is always positive**, largely because of the small value of $\bar\tau$. We plot the changing of these terms on the ResNet18 experiment with the table and figure in the rebuttal pdf. **The order of difference** is usually smaller than 1e-8 and approaches the order of 1e-12 in the end.
>
> - **The evaluation metric in Figure 2.** We want to show our methods can show a better robustness and accuracy tradeoff for both PGD and AutoAttack for a more comprehensive evaluation. As shown in Figure 3, gradient projection also shows a better robustness accuracy tradeoff evaluated by AutoAttack.
>
> - **The difference between MAX and RAMP with lambda=0**. RAMP with $\lambda = 0$ and not considering the key tradeoff pair will become the same as MAX. Therefore, MAX is a special case of RAMP.
>
> [1] https://arxiv.org/abs/1901.08573
>
> [2] https://openreview.net/forum?id=rklOg6EFwS
>
> [3] https://openreview.net/forum?id=Azh9QBQ4tR7
>
> [4] https://arxiv.org/abs/2302.05049
>
> [5] https://arxiv.org/abs/2106.09129

---

> > ### Author Response · Authors · 2024-08-09
> > **follow up**
> >
> > Dear reviewer,
> >
> > We want to thank you again for your efforts in reviewing our submission.
> >
> > We have tried our best to respond to your questions. If our responses address your concerns, we sincerely hope you can reconsider the scores. We will also be very happy to answer any follow-up questions. Look forward to your updates.
> >
> > Thanks,
> > Authors

---

> > ### Comment · Reviewer_3Hj9 · 2024-08-13
> >
> > I thank the authors for the detailed responses.
> >
> > 1. Regarding the perturbation beyond $l_1$, $l_2$ and $l_\infty$ attacks, I believe including the tables presented in the rebuttal is beneficial. However, the authors should also provide detailed information about the implementation of $l_0$ attack, common corruption attack e.t.c. for readers to comprehensively assess the effectiveness of the proposed method. This needs a significant revision of the paper, and I believe it needs a re-submission to ensure a rigorous review process.
> >
> > 2. I agree with Reviewer 6TV5 that the correlation among the distributions caused by different types of perturbation is not clearly investigated in the theoretial analyses. In addition, the algorithmic difference between the proposed method and existing ones (such as TRADES) is not big. Therefore, I have concerns about the novelty of this work.
> >
> > Based on the issues above, I regret to say that I cannot increase my rating based on the current manuscript.

---

> > > ### Author Response · Authors · 2024-08-13
> > > **Thank you and we provide a further explanation**
> > >
> > > Thanks for your reply and here we provide a further explanation.
> > >
> > > **Regarding the implementation**: For l0 attack, we use [1] with an epsilon of 9 pixels and 5k query points. For common corruptions, we directly use the implementation of robustbench [2] for evaluation across 5 severity levels on all corruption types used in [3]. For other unseen adversaries, we follow the implementation of [4], where we set eps=12 for fog attack, eps = 0.5 for snow attack, eps=60 for gabor attack, eps=0.125 for elastic attack, and eps=0.125 for jpeg linf attack (we have mentioned those epsilons in the previous table). We will update the implementation information in the next version of our paper.
> > >
> > > We think it is standard to provide additional results and respond to reviewer requests at NeurIPS (see examples [4,5]), which does not necessarily require a resubmission.
> > >
> > > **Correlations among the distributions**. Please see our newest response to Reviewer 6TV5.
> > >
> > > [1] https://github.com/fra31/sparse-rs/tree/master
> > >
> > > [2] https://github.com/RobustBench/robustbench/tree/master/robustbench
> > >
> > > [3] Benchmarking Neural Network Robustness to Common Corruptions and Perturbations.
> > >
> > > [4] https://github.com/cassidylaidlaw/perceptual-advex
> > >
> > > [5] https://openreview.net/forum?id=IkD1EWFF8c
> > >
> > > [6] https://openreview.net/forum?id=v5Aaxk4sSy

---

### Official Review · Reviewer_6TV5 · 2024-07-08

**Soundness:** 2
**Presentation:** 3
**Contribution:** 2
**Rating:** 5
**Confidence:** 3

**Summary:**

This work proposes a training framework RAMP, to boost the robustness against multiple $\ell_p$ perturbations, through connecting NT with AT via gradient projection.

**Strengths:**

- Easy to follow and clear presentation
- Enough details
- Some theoretical analyses
- Outwardly, this is a good paper

**Weaknesses:**

My main concern is why the distributions of adversarial examples generated using $\ell_1$, $\ell_2$, and $\ell_\infty$ norms are distinct?
Is this due to inconsistent (unreasonable) radius settings across different $\ell_p$-norms?
What constitutes a reasonable radius setting for various norms?
These norms can be bounded by each other theoretically, what is the essential meaning to study this problem?

**Questions:**

ref weakness

**Limitations:**

As I mentioned before, in my opinion, I think this work needs to discuss more about the essential meaning of the problem.

---

> ### Author Rebuttal · Authors · 2024-08-06
>
> - **The essential meaning of studying this problem:**
>
> a) In real-world settings, adversarial attacks often involve multiple types of perturbations, necessitating models to be robust against $l_1$, $l_2$, and $l_\infty$ attacks. For instance, in image recognition: $l_1$ attacks modify key pixels to bypass facial recognition; $l_2$ attacks apply slight blur or noise to fool object detection; $l_\infty$ attacks adjust color balance or contrast to cause misclassification. These examples highlight the need for comprehensive adversarial training. Robustness against multiple $l_p$ perturbations is critical, as it mirrors real-world scenarios where adversaries use a blend of techniques, ensuring the security and reliability of machine learning systems.
>
> b) Training for multiple norm robustness allows generalization beyond the threat model used during training [1]. Our method achieves superior average and union accuracy compared to the SOTA common corruption model [7], $l_p$ pretrained models, E-AT, and MAX across other perturbations beyond multiple-norm perturbations. Specifically, it obtains good accuracy against common corruptions with five severity levels and shows the best average and union accuracy performance against multiple unseen adversaries. Using WideResNet 28-10 models, we observe the following results:
>
> For common corruptions, RAMP achieves a 4% improvement compared to E-AT and MAX, averaged across five severity levels and all common corruptions.
>
> | model           | common corruptions |
> |-----------------|--------------------|
> | l1-AT           |               78.2 |
> | l2-AT           |               77.2 |
> | linf-AT         |               73.4 |
> | winninghand [7]  |           **91.1** |
> | E-AT  |               71.5 |
> | MAX |                 71 |
> | RAMP |               75.5 |
>
> For L0 and other unseen adversaries, RAMP shows the best average and union accuracy, with a 4% improvement over E-AT and a 2% improvement over MAX.
>
> | model           | l0 (eps=9,query=5k) | fog (eps=128) | snow (eps=0.5) | gabor (eps=60) |  elastic (eps=0.125) | jpeg(eps=0.125) | avg      | union    |
> |-----------------|---------------------|---------------|----------------|----------------|----------------------|-----------------|----------|----------|
> | l1-AT           |                  79 |          41.4 |           22.9 |           40.5 |                 48.9 |            48.4 |     46.9 |     12.8 |
> | l2-AT           |                67.5 |          48.7 |           26.1 |           44.1 |                 53.2 |            45.4 |     47.5 |     16.2 |
> | linf-AT         |                55.5 |          44.7 |           32.9 |           53.8 |                 56.6 |            33.4 |     46.2 |     19.1 |
> | winninghand [7]  |                74.1 |          74.5 |           18.3 |           76.5 |                 12.6 |               0 |     42.7 |        0 |
> | E-AT  |                58.5 |          35.9 |           35.3 |           50.7 |                 55.7 |            60.3 |     49.4 |     21.9 |
> | MAX |                56.2 |          42.9 |           35.4 |           49.8 |                 57.8 |            55.7 |     49.6 |     24.4 |
> | RAMP |                55.5 |          40.5 |           40.2 |           52.9 |                 60.3 |            56.1 | **50.9** | **26.1** |
>
>
> These results highlight the effectiveness of our method in enhancing model robustness across diverse adversarial scenarios.
>
> - **Why are distributions distinct?** The distributions are distinct because these norms measure distances differently, resulting in different adversarial example distributions. Successful defenses are often tailored to specific perturbation types, providing empirical or certifiable robustness guarantees for those types alone. These defenses typically fail to offer protection against other attacks [1]. Moreover, enhancing robustness against one perturbation type can sometimes increase vulnerability to others [4,5], which further confirms the distinction between multiple-norm distributions.
>
> - **What constitutes a reasonable radius setting?** The values we are using are standard in the previous literature [1,2,3]; the epsilon values are chosen so that input images can be adversarially perturbed to be misclassified but are “close enough” to the original example to be imperceptible to the human eye [3]. Also, in Table 2 and Appendix Tables 5-16, we show RAMP works not only for the standard choice of epsilons but varying values of epsilons. The reasonable radius setting for various norms should make each pair of two perturbations not include one another. For given epsilon values of $\epsilon_1$, $\epsilon_2$, $\epsilon_\infty$ and $d$ (dimension of the inputs), we need the following relationships to achieve this:
> $$0 \leq \epsilon_1 \leq d \cdot \epsilon_\infty$$
> $$0 \leq \epsilon_2 \leq \sqrt{d} \cdot \epsilon_\infty$$
> $$\frac{\epsilon_1}{\sqrt{d}} \leq \epsilon_2 \leq \epsilon_1$$
>
> We check the standard choices of CIFAR10 $\epsilon_1=12, \epsilon_2=0.5, \epsilon_\infty=\frac{8}{255}, d = 3 \cdot 32^2$ and Imagenet $\epsilon_1=255, \epsilon_2=2, \epsilon_\infty=\frac{4}{255}, d = 3 \cdot 224^2$. For CIFAR10, we have $0 \leq 12 \leq 3072\cdot\frac{8}{255}, 0 \leq 0.5 \leq \sqrt{3072}\cdot\frac{8}{255}, \frac{12}{\sqrt{3072}} \leq 0.5 \leq 12$. For Imagenet, we have $0 \leq 255 \leq 150528\cdot\frac{4}{255}, 0 \leq 2 \leq \sqrt{150528}\cdot\frac{4}{255}, \frac{255}{\sqrt{150528}} \leq 2 \leq 255$. Both hold so our choices make three perturbations not include one another. However, **the key tradeoff pair always exists**. The two perturbations with the largest volumes refer to the strongest attack as the attacker has more search area.
>
> [1] https://arxiv.org/abs/2105.12508
>
> [2] https://arxiv.org/abs/1904.13000
>
> [3] https://arxiv.org/abs/1909.04068
>
> [4] https://openreview.net/forum?id=BJfvknCqFQ
>
> [5] https://arxiv.org/abs/1805.09190
>
> [6] https://arxiv.org/abs/1903.12261
>
> [7] https://arxiv.org/abs/2106.09129

---

> > ### Author Response · Authors · 2024-08-09
> > **follow up**
> >
> > Dear reviewer,
> >
> > We want to thank you again for your efforts in reviewing our submission.
> >
> > We have tried our best to respond to your questions. If our responses address your concerns, we sincerely hope you can reconsider the scores. We will also be very happy to answer any follow-up questions. Look forward to your updates.
> >
> > Thanks,
> > Authors

---

> ### Comment · Reviewer_6TV5 · 2024-08-10
> **response to author**
>
> I thank authors for their detailed response.
>
> **"In real-world settings, adversarial attacks often involve multiple types of perturbations, necessitating models to be robust against $\ell_1$, $\ell_2$, and $\ell_\infty$ attacks."**
> There are so many vector norms and matrix norms, why $\ell_1$, $\ell_2$, and $\ell_\infty$ vector norms are important in **real-world** adversarial attacks?
> While this work combines previously established norm-based attacks, it appears to be an incremental advancement rather than a fundamental exploration.
> The crucial question remains unanswered: What intrinsic properties of these three norms make them more suitable for **mixture** compared to other potential norms? A deeper investigation into this fundamental issue could provide valuable insights and potentially lead to more robust defense mechanisms.
>
> **"The distributions are distinct because these norms measure distances differently"**
> Given that these norms can be bounded by one another, I believe it is more accurate to characterize their relationship as ''correlated'' rather than ''distinct''. Further exploration of this correlation could yield valuable insights.
>
> **"$\epsilon_1$, $\epsilon_2$, $\epsilon_\infty$ in experiments"**
> While the corner cases (or extremal spaces) for different norms are indeed distinct, a critical question remains: What justifies the focus on these three specific norms ($\ell_1$, $\ell_2$, and $\ell_\infty$) rather than exploring corner cases for other norms?
>
> All in all, while this work contributes to the field, it appears to be an incremental advancement. It would be better to explore the above questions. I sit on the fence for this work.

---

> > ### Comment · Reviewer_6TV5 · 2024-08-12
> > **response**
> >
> > To avoid misleading the AC, I increase my score to 5, but I sit on the fence for this work.

---

> ### Author Response · Authors · 2024-08-12
> **Thank you and more justifications on why l1, l2, linf threat models are important for real-world applications.**
>
> We thank the reviewer for the detailed reply and here we provide further justifications on why $l_1$, $l_2$, $l_\infty$ attacks are important for real-world attacks.
>
> 1. We would like to emphasize that the use of  $l_1$, $l_2$, and $l_\infty$ norms has been extensively adopted in adversarial learning research, primarily as a foundational approach to enhance robustness against well-defined and mathematically tractable threat models [1,2,3]. While these attack models may appear straightforward, the challenge of developing models that are genuinely robust against them at the same time remains a significant and unresolved issue. The critical importance of these simpler attack paradigms is further underscored by their central role in the most widely used adversarial toolkits, such as Foolbox [4], and in benchmarks like RobustBench [5], which predominantly focus on evaluating robustness against these norm-based attacks. Further, many physical real-world adversarial attacks [7,8,9,10] are based on these norms.
>
> 2. RAMP shows a better ability to generalize to more complex norms. For example, in the case of the Wasserstein adversarial attack [11], RAMP outperforms E-AT and MAX by 1-3% across different perturbation sizes, as demonstrated in the table below. Besides, we are happy to try other norms suggested by the reviewer if there are any.
>
> |      |  eps = 0.001 |  eps = 0.003 |  eps = 0.005 |
> |------|--------------|--------------|--------------|
> | MAX  |         75.7 |         70.6 |         66.1 |
> | E-AT |         76.6 |         71.8 |     67.7 |
> | RAMP |     79.5 |     73.4 |     67.5 |
>
> 3. Techniques for enhancing  $l_1$,  $l_2$, and $l_\infty$ robustness can be extended to address perturbation sets that are not easily defined mathematically, such as image corruptions or adversarial lighting variations [6]. Our new results on generalizing to other perturbations and corruptions (see tables in response to reviewers 6TV5, 3Hj9, and Qhq5) confirm that point with the best robustness-accuracy tradeoff among $l_p$ pretrained, SOTA image corruption, E-AT, and MAX models.
>
> In conclusion, RAMP represents an advancement in enhancing union adversarial robustness and the robustness-accuracy tradeoff across norms/perturbations/corruptions, while maintaining a relatively good computational efficiency.
>
>
>
>
> [1] Explaining and harnessing adversarial examples.
>
> [2] Towards evaluating the robustness of neural networks.
>
> [3] Towards Deep Learning Models Resistant to Adversarial Attacks.
>
> [4] https://github.com/bethgelab/foolbox/tree/master
>
> [5] https://robustbench.github.io/
>
> [6] Learning perturbation sets for robust machine learning.
>
> [7] Accessorize to a Crime: Real and Stealthy Attacks on State-of-the-Art Face Recognition.
>
> [8] Robust Physical-World Attacks on Deep Learning Visual Classification.
>
> [9] Physical Adversarial Examples for Object Detectors.
>
> [10] Synthesizing Robust Adversarial Examples.
>
> [11] Stronger and Faster Wasserstein Adversarial Attacks.

---

> > ### Comment · Reviewer_6TV5 · 2024-08-13
> > **response**
> >
> > Authors also mentioned $\ell_{1,2,\infty}$ are commmonly used in adversarial robustness research, thus they studied the mixture of them. Combining these norms seems more like an incremental work. The real question is:
> > **What's the correlation between adversarial distributions generated by different norms? How RAMP deals with this correlation?** This work did not provide any theoretical analysis here.
> >
> > In this point, I agree with Reviewer 3Hj9, this work lacks novelty, just an incremental work.

---

> > > ### Author Response · Authors · 2024-08-13
> > > **Further response on correlations among adversarial distributions**
> > >
> > > Thank you for your further questions and we respond to them as follows.
> > >
> > > People have tried to theoretically characterize the adversarial distribution [1,2,3], but to the best of our knowledge, there have been no successful attempts at the precise characterizations of adversarial distributions for different norms theoretically. This is because the adversarial distributions change dynamically during training due to updates to model weights. Even after many years, people have not been able to characterize it for a single norm on DNNs. We believe that this problem itself is challenging and deserves a separate investigation.
> > >
> > > Our goal is to achieve better robustness-accuracy tradeoff by providing insights into the relationship between natural and adversarial distributions (Theorem 4.6 and 4.7 in our paper). Our theory does not explicitly analyze the correlations among different norms, as we approach them collectively by treating the mixture as a single distribution. However, we offer insights into robustness-accuracy tradeoff under this framework, which applies to both individual and multiple attacks. This perspective provides a generalizable understanding that is relevant across various attack scenarios.
> > >
> > > Further, we studied the correlations between adversarial distributions of multiple norms by identifying the key tradeoff pair. The distributions generated by the two strongest attacks show the largest shifts from the natural distribution; also, they have the largest distribution shifts between each other because of larger and most distinct search areas. Thus, by calculating the ball volume for each attack, we select the two with the largest volumes as the key tradeoff pair, which also informs our logit pairing method. Our current approach demonstrates effectiveness by improving union accuracy with significant margins.
> > >
> > > Also, we want to point out that we study the multiple-norm robustness problem not only because $l_1, l_2, l_\infty$ norms are commonly used. We show RAMP with adversarial training on these norms can generalize to a variety of other perturbations and corruptions, serving as the basis for many real-world attacks. This is supported by several results we provided in our previous rebuttal, illustrating their applicability and relevance in practical scenarios.
> > >
> > > [1] Explicit Tradeoffs between Adversarial and Natural Distributional Robustness.
> > >
> > > [2] Precise Tradeoffs in Adversarial Training for Linear Regression.
> > >
> > > [3] Fundamental Tradeoffs in Distributionally Adversarial Training.

---

### Official Review · Reviewer_jXrr · 2024-07-11

**Soundness:** 3
**Presentation:** 2
**Contribution:** 3
**Rating:** 7
**Confidence:** 4

**Summary:**

The paper introduces a new addition to the adversarial training literature -- it proposes to augment standard adversarial training [Madry et al.] with a specific logits pairing loss and, the existing technique, of gradient project.

The paper shows strong results on the union of lp-norm adversarial accuracy (over l-1, l-inf and l-2 using the standard CIFAR-10 dataset and standard perturbation radii).

**Strengths:**

The paper describes its contributions and motivates why each technique is introduced. The paper includes an informal discussion to motivate the logits pairing loss and some supporting theoretical analysis.

The paper obtains strong results (against many relevant baselines, evaluated correctly with AutoAttack) -- which is rare for adversarial robustness submissions. The method also seems straightforward to implement. Happy to recommend acceptance, based mainly on the strong results.

**Weaknesses:**

The one main weakness is that Table 1 must be re-generated using all the test set points in CIFAR-10, not just 1000.

Presentation-wise, Figure 1 is very confusing; there is no need to plot t-SNEs of image when one wants to simply show classification labels. Please consider re-plotting this figure (with a line plot, histogram, etc.).

The writing, overall, is ok, but could also be improved: for example, lines 162-167, are approximately repeated from earlier on in the paper.

**Questions:**

See above.

**Limitations:**

The authors have adequately addressed the limitations of their work.

---

> ### Author Rebuttal · Authors · 2024-08-06
>
> - *The one main weakness is that Table 1 must be re-generated using all the test set points in CIFAR-10, not just 1000.*
>
> Thank you for the valuable suggestion. We have re-evaluated both RAMP and E-AT using the entire CIFAR-10 dataset. The results demonstrate that RAMP consistently enhances union accuracy compared to E-AT (up to 2.2%), corroborating the trend observed in our initial findings. We will update these numbers in the revised version of the paper.
>
> | Models               | Methods | Clean | linf | l2   | l1   | union    |
> |----------------------|---------|-------|------|------|------|----------|
> | RN-50                | E-AT    | 85.3  | 46.5 | 68.3 | 45.3 | 41.6     |
> |                      | RAMP    | 83.9  | 47.3 | 67.4 | 47.1 | **43.8** |
> | WRN-34-20            | E-AT    | 87.8  | 49.0 | 71.6 | 49.8 | 45.1     |
> |                      | RAMP    | 87.0  | 49.8 | 70.7 | 50.8 | **47.3** |
> | WRN-28-10 (*) carmon | E-AT    | 89.3  | 51.8 | 74.6 | 53.3 | 47.9     |
> |                      | RAMP    | 88.7  | 52.3 | 73.1 | 51.7 | **48.8** |
> | WRN-28-10 (*) gowal  | E-AT    | 89.8  | 54.4 | 76.1 | 56.0 | 50.5     |
> |                      | RAMP    | 89.1  | 55.1 | 74.5 | 54.8 | **52.1** |
> | WRN-70-16 (*)        | E-AT    | 89.6  | 54.4 | 76.7 | 58.0 | 51.6     |
> |                      | RAMP    | 90.1  | 55.1 | 74.5 | 57.4 | **53.4** |
>
> - *Presentation-wise, Figure 1 is very confusing; there is no need to plot the t-SNEs of the image when one wants to show classification labels simply. Please consider re-plotting this figure (with a line plot, histogram, etc.). The writing, overall, is ok, but could also be improved: for example, lines 162-167, are approximately repeated from earlier on in the paper.*
>
> Thanks for the suggestions on presentation and writing. We replot Figure 1 using histograms on accuracy (included in the rebuttal pdf), using CIFAR-10 training data: the x-axis represents the robustness against different attacks and the y-axis is the accuracy. The figure shows that after 1 epoch of finetuning on $l_1$ examples or performing EAT, we lose much $l_\infty$ robustness since blue/yellow histograms are much lower than the red histogram under the Linf category. In comparison, RAMP preserves both $l_\infty$ and union robustness more effectively compared to E-AT and fine-tuning on $l_1$ examples; the green histogram is higher than the red/yellow histogram under Linf and Union categories. Specifically, RAMP maintains 14%, and 28% more union robustness than E-AT and $l_1$ fine-tuning. Besides, we will make writing more concise in the updated version.

---

> > ### Author Response · Authors · 2024-08-09
> > **follow up**
> >
> > Dear reviewer,
> >
> > We want to thank you again for your efforts in reviewing our submission.
> >
> > We have tried our best to respond to your questions. If our responses address your concerns, we sincerely hope you can reconsider the scores. We will also be very happy to answer any follow-up questions. Look forward to your updates.
> >
> > Thanks,
> > Authors

---

### Author Rebuttal · Authors · 2024-08-06

We would like to thank the reviewers for their valuable feedback and constructive suggestions. We hope our responses have adequately addressed the questions and concerns raised by the reviewers. Here we summarize our response to the main concerns of the reviewers.

- **The essential meaning of this problem** (Reviewer 6TV5)

**Enhancing robustness against one type is not enough**: Successful defenses are often tailored to specific perturbation types, providing empirical or certifiable robustness guarantees for those types alone. These defenses typically fail to offer protection against other attacks [1]. Moreover, enhancing robustness against one perturbation type can sometimes increase vulnerability to others [4,5].

**The choices of epsilon values are standard and comprehensive**: The values we are using are standard in the previous literature [1,2,3]; the epsilon values are chosen so that input images can be adversarially perturbed to be misclassified but are “close enough” to the original example to be imperceptible to the human eye [3]. Also, in Table 2 and Appendix Table 5-16, we show that RAMP works not only for the standard choice of epsilons but also for the varying values of epsilons.

**Training a model to be robust against multiple $l_p$ perturbations is crucial as it mirrors real-world scenarios where adversaries use a blend of techniques**. In image recognition, an $l_1$ attack might modify key pixels to bypass facial recognition, an l_2 attack could apply slight blur or noise to fool an object detection system, and an $l_\infty$ attack might adjust the color balance or contrast to cause misclassification. This holistic approach enhances the resilience, security, and reliability of machine learning systems in practical applications.

- **The novelty and generality of RAMP** (Reviewer 3Hj9 and 6TV5)

**Logit pairing is not the same as TRADES [6]**: $p_r$ represents the predictions of $l_r$ adversarial examples, not the clean input probability. Our logit pairing method differs from TRADES by regularizing only the logits on the correctly predicted $l_r $adversarial subset, enhancing union accuracy by aligning $l_r$ and $l_q$ attack logit distributions. Combining RAMP with TRADES loss improves union accuracy to 45.2% from 44.6% in the WideResNet-28-10 experiment.

**New findings on accuracy and robustness tradeoff**: Our gradient projection method generalizes better than traditional methods like TRADES [6], MART [7], and HAT [8], which focus on specific perturbations. Our approach is effective for both single and other various types of perturbations, offering improved robustness-accuracy tradeoff.

- **Generality of RAMP**: Training for multiple norm robustness allows generalization beyond the threat model used during training [1]. Our method achieves superior average and union accuracy compared to $l_p$ pretrained models, [9], E-AT, and MAX across other perturbations beyond multiple-norm perturbations. Compared with MAX and E-AT, it excels 4% in accuracy against common corruptions with five severity levels and shows 2-4% better union accuracy against multiple unseen adversaries on the WideResNet experiment (Tables in response to reviewer 3Hj9), showing our method’s robustness and adaptability to diverse adversarial scenarios.

- **Robustness-accuracy tradeoff** (Reviewer Qhq5): It is possible to use RAMP to obtain an accuracy close to E-AT and a better union accuracy by setting a small $\lambda$. For instance, in Figure 4(b), when $\lambda=1$, RAMP has a clean accuracy of 81.9% and union accuracy of **44.4%**, compared with E-AT's 82.2% and 42.4%. Further, RAMP obtains a better robustness-accuracy tradeoff by generalizing better to common corruptions and other unseen adversaries, and the overall gain in robustness is more than the drop in accuracy. Compared with E-AT in the WRN-28-10 and RN-18 experiments on CIFAR-10, RAMP shows a better tradeoff between robustness and accuracy against various kinds of perturbations and corruptions (Tables in response to reviewer Qhq5).

- **New time complexity results, generated results of Table 1 using all test points, regenerated Figure 1** (Reviewer Qhq5 and jXrr).

a) We present additional time cost results on CIFAR-10 and Imagenet for wideresnet and transformer experiments, supporting that RAMP is more expensive than E-AT and less expensive than MAX.

b) We have re-evaluated both RAMP and E-AT using the entire CIFAR-10 dataset (see Table in response to reviewer jXrr). RAMP consistently enhances union accuracy compared to E-AT (up to 2.2%), corroborating the trend observed in our initial findings. Also, we observe that the gain in the union accuracy is usually larger than the loss in the clean accuracy.

c) We replot Figure 1 using histograms on accuracy (included in the rebuttal pdf): RAMP maintains 14%, and 28% more union robustness than E-AT and $l_1$ fine-tuning after 1 epoch of robust fine-tuning on CIFAR-10 training data.

- **Novelty and precision of the theory** (Reviewer 3Hj9) : The novelty of Theorem 4.5 lies in its consideration of the changing adversarial example distribution $\mathcal{D}_{a^t}$ at each time step $t$ to get the convergence proof. Theorem 4.5 links the convergence of the aggregation rule with the delta error analysis in Theorem 4.6, showing that a smaller delta error leads to better convergence with a tighter bound. Further, we updated Theorem 4.6 and plotted the changing of delta error differences in the rebuttal PDF to show GP always has a smaller delta error than standard adversarial training, which supports the experimental results.

[1] https://arxiv.org/abs/2105.12508

[2] https://arxiv.org/abs/1904.13000

[3] https://arxiv.org/abs/1909.04068

[4] https://openreview.net/forum?id=BJfvknCqFQ

[5] https://arxiv.org/abs/1805.09190

[6] https://arxiv.org/abs/1901.08573

[7] https://openreview.net/forum?id=rklOg6EFwS

[8] https://openreview.net/forum?id=Azh9QBQ4tR7

[9] https://arxiv.org/abs/2106.09129

---

### Decision · Program_Chairs · 2024-09-25

**Decision:**

Accept (poster)

**Comment:**

This paper studies the question of jointly achieving robustness against multiple adversarial perturbations. The main idea is to use a logit-pairing loss that makes the logits achieved on multiple perturbations match each other. Then, the authors used gradient projection to further align the updates from adversarial examples and ordinary non-adversarial examples.

Two reviewers are concerned about the paper. For example, one reviewer believes that logit pairing sounds like a “standard” idea and lacks novelty. Furthermore, focusing on the L1, L2, and L-inf attacks are not representative enough, and more studies on other types of perturbations should be considered, such as common corruptions, which the authors have addressed during the rebuttal. There is another concern that the paper does not provide precise characterizations of adversarial distributions for different norms.

Two other reviewers favored accepting the paper due to the solid theoretical and empirical results. One reviewer even championed the paper despite the diverging scores.

IMHO, the problem of studying the tradeoff between different norms is legitimate because that’s a good starting point for the community to go beyond a single perturbation norm. For example, it’s quite intuitive that L2 and L-infinity attacks will lead to different distributions of adversarial examples, and training against one type of perturbation can lead to a distributional shift on the other. I think this is a common problem, and focusing on the L1, L2, and L-infinity is probably due to the convention in this field and the consideration of tractability.

The reviewers provided some good suggestions for improving the paper. For example, instead of showing a T-SNE plot in Figure 1, one can just show a line plot. After discussing with the reviewer, the authors showed that RAMP achieves an improved accuracy-robustness tradeoff than a previous baseline, which convinced one reviewer.

In summary, I recommend accepting the paper despite the diverging scores.